# Discovery of diverse anellovirus sequences in Thai human sequencing data

Worakorn Phumiphanjarphak,[1,2] Jinjutha Parkbhorn,[2] Chumpol Ngamphiw,[3] Sissades Tongsima,[3] Pakorn Aiewsakun[1,2]

**ABSTRACT** Anelloviruses are part of the normal human viral flora. Although their diversity in humans has been investigated in many countries, and despite their initial detection in Thailand in 1999, knowledge of Thai anelloviruses remains very limited. This study analyzed 1,175 whole-genome sequencing data sets from Thai individuals to mine for potential anellovirus sequences. Our analyses detected anellovirus sequences in 149 data sets (12.68%), uncovering 434 partial anellovirus sequences and 77 complete genome sequences, characterized by the presence of terminal redundancy, complete *orf1*, and the conserved untranslated region upstream of the *orf1* gene. Sequence analyses indicated that these viruses belong to seven genera, including *Alphatorquevirus*, *Betatorquevirus*, *Gammatorquevirus*, *Hetorquevirus*, *Lamedtorquevirus*, *Samektorquevirus*, and *Yodtorquevirus*. Notably, *Hetorquevirus*, *Lamedtorquevirus*, *Samektorquevirus*, and *Yodtorquevirus* had not previously been reported in Thailand. Phylogenetic analysis of ORF1 protein sequences showed that Thai anelloviruses form multiple phylogenetic clusters with non-Thai anelloviruses, indicating frequent cross-country transmission and multiple origins of the virus in Thailand. Furthermore, sequence similarity network analysis identified 33 potentially novel anellovirus species in our data set. Our findings greatly expand the knowledge of anellovirus diversity in Thailand and demonstrate the potential of human whole-genome sequencing data as a valuable resource for viral discovery. Lastly, we highlight and discuss some challenges with the use of the current pairwise sequence similarity-based classification scheme, in particular, how gaps can influence similarity calculation and potentially lead to inconsistencies with a phylogenetic-based classification scheme.

**IMPORTANCE** Anelloviruses are widespread in humans, yet their diversity remains poorly characterized in many regions, including Thailand. Here, we demonstrate that human sequencing data sets, originally generated without the intention for virome research, can be effectively mined for anellovirus sequences, including complete genomes. Our findings reveal a substantial number of previously unreported anelloviruses in Thailand, significantly expanding the known diversity of the virus. We also highlight potential limitations of the current anellovirus species classification scheme, which is based on pairwise *orf1* sequence similarity analysis with a hard threshold cutoff at 69%. Our results reveal that the current scheme can sometimes yield taxonomic groupings that are inconsistent with phylogenetic relationships, particularly when significant alignment gaps are present. Overall, our results show that existing human sequencing data can be effectively repurposed for virus discovery research and suggest the need for more robust and phylogenetically informed classification frameworks as viral sequence databases continue to expand.

**KEYWORDS** anellovirus, *Anelloviridae*, virus discovery, virome

**Peer Reviewer** Chutchai Piewbang, Chulalongkorn University Faculty of Veterinary Science, Bangkok, Thailand

Address correspondence to Pakorn Aiewsakun, pakorn.aie@mahidol.ac.th.

The authors declare no conflict of interest.

See the funding table on p. 19.

**1**

Anelloviruses are a diverse group of negative-sense single-stranded DNA viruses with small circular genomes (~2–4 kilobases), containing 3 to 5 open reading frames (ORFs) (1–3). They can infect a wide range of mammals (2, 4) and are recognized as endemic to humans (3). As of August 2024, the International Committee on Taxonomy of Viruses (ICTV) recognizes 173 species and 34 genera of anelloviruses (5). At present, anellovirus species are defined based on pairwise sequence similarity analysis of *orf1*—the largest gene in the virus genomes that can be found in all anellovirus subgroups encoding the virus capsid protein—with the similarity cutoff set at 69% (4).

Although anellovirus diversity in humans has been investigated in many countries across the globe, the knowledge about these viruses in Thailand remains very limited. Among the first studies of anelloviruses in Thailand was conducted by Tangkijvanich et al. in 1999 (6), which investigated the prevalence of alphatorqueviruses (formerly known as torque teno viruses) in humans primarily by using serological tests and polymerase chain reaction (PCR) assays. Between 2000 and 2004, a number of studies reported anellovirus prevalence in Thailand in various hosts, including humans (7–9), gibbons (10), and pigs (11) by PCR. With high-throughput sequencing (HTS) technologies, a number of studies from Thailand recently reported these viruses from humans (12–14) and macaques (15). Nonetheless, thus far, anelloviruses detected in Thailand have only been reported at the sequence read level, with no complete assembled genomes published.

In this study, we report the identification of anellovirus sequences mined from 1,175 HTS data sets derived from Thai human individuals. Our analysis uncovered 434 partial sequences and the first 77 complete genome sequences of the virus from Thailand from 149 samples (12.68%), spanning across seven genera, including *Alphatorquevirus*, *Betatorquevirus*, *Gammatorquevirus*, *Hetorquevirus*, *Lamedtorquevirus*, *Samektorquevirus*, and *Yodtorquevirus*. Notably, the latter four are reported in Thailand for the first time, and our analyses also identified a total of 33 potentially novel virus species in our data set. These findings significantly improve our knowledge of anellovirus diversity in Thailand. Additionally, challenges with the use of the current pairwise sequence similarity analysis for anellovirus species demarcation are discussed.

## MATERIALS AND METHODS

### Human whole-genome sequencing data

All sequence data sets analyzed in this study were from an in-house database maintained by the National Center for Genetic Engineering and Biotechnology (BIOTEC), Thailand. All were paired-end short reads generated by HTS technologies, with human reads removed through read-mapping against the reference human genome hg38.

### Detection of anellovirus sequences

We used the Entourage pipeline (16) to detect anellovirus sequences. First, the read assembly module was used to clean and assemble reads into contigs. Briefly, the pipeline trimmed sequencing adapters and removed low-quality reads by using fastp version 0.21 (17) with the following parameters "-q 25 -u 20 -n 10 -e 30 -L 50"; i.e., reads were removed if (i) more than 20% of their bases had Phred quality scores <25; (ii) they contained more than 10 "N" bases; (iii) they had an average base quality score <30; or (iv) they were shorter than 50 bases after cleaning. High-quality reads were then *de novo* assembled into contigs using MEGAHIT version 1.2.9 (18) with the parameters "--k-min 21 --k-max 121 --k-step 10," while other settings were left as default.

The assembled contigs were subsequently analyzed by the Entourage discovery module to detect the presence of potential anellovirus sequences. This module first assigned a taxonomic group to each contig by identifying potential protein-coding regions using MMseqs2 version 13.45111 (19). Assembly quality of the putative viral contig was then assessed with CheckV version 1.0.1 (20) with the database version 1.5. To create the MMseqs2 database, we downloaded 32,087 publicly available anellovirus

protein sequences (family *Anelloviridae*) from the NCBI Protein database (access date: 10 July 2023). These sequences were clustered based on protein sequence similarity (≥90%) using CD-HIT version 4.8.1 (21), resulting in 5,786 clusters. The longest sequence from each cluster was selected as the representative to construct the MMseqs2 database.

Putative anellovirus sequences identified by the discovery module underwent further validation through reciprocal BLASTN or BLASTP searches using NCBI BLAST+ version 2.15.0 (22). First, sequences were queried against the NCBI nt database using BLASTN with an e-value threshold of $1 \times 10^{-30}$. Sequences with anellovirus BLASTN best hits were classified as true positives and retained for further analysis, while those with non-anellovirus BLASTN best hits were excluded. For sequences without BLASTN hits returned, protein sequences predicted from ORFs with lengths ≥300 bases were extracted and queried against the NCBI nr database by using BLASTP with an e-value threshold of $1 \times 10^{-10}$. Sequences with anellovirus BLASTP best hits were also classified as true positives and retained. Sequences without any anellovirus sequence BLASTN/P best hits were excluded from subsequent analysis.

This process was iteratively performed until no new sequences could be identified. In each iteration, the reference MMseqs2 database was updated to include protein sequences from newly identified anelloviruses that met a minimum length threshold of 100 amino acids.

## Anellovirus genome scaffolding

Clean reads were mapped back to the identified anellovirus sequences using BWA-MEM2 version 2.2.1 (23) with default settings. The resulting read alignments were visually inspected using IGV version 2.17.4 (24). Anellovirus contigs were manually scaffolded based on paired-end read linkages deduced from the read mapping information. In order to avoid generating *in silico* chimeric sequences as much as our data allowed, two contigs were linked together only when their scaffolding was supported solely by "proper" read pairs (i.e., mapped paired-end reads showing the correct "inward" orientation, →←, and they were separated by a reasonable distance expected based on the library preparation or other read pairs), and all supporting read pairs must have also supported only such the scaffolding, and not any other scaffoldings; otherwise, the contigs would have been left unscaffolded. Potential chimeric sequences among scaffolds were further checked by searching for "split" reads in the back-mapping read alignments (i.e., individual reads that could be split into multiple subsegments and mapped to multiple non-contiguous regions) by using SAMtools (25), but no split reads were detected. For scaffolds containing sequence gaps, gap sizes were estimated by aligning the scaffolds to their closest matches in the NCBI nt database. Distances between read pairs supporting our manual scaffoldings were computed and compared to those of other read pairs in the same data sets. None were found to have unexpected distances, supporting that all read pairs supporting scaffoldings were indeed proper read pairs.

After obtaining scaffolds, consensus sequences were generated using iVar version 1.4.2 (26) with the "-t 0" option, which calls the most frequent base at each position as the consensus base. Assembly quality was assessed again using CheckV version 1.0.1 (20) with the database version 1.5, and, finally, genome sequence completeness was determined based on the presence of terminal redundancy, complete *orf1*, and the conserved untranslated region (UTR) located upstream of the *orf1* gene.

## Reference *orf1* database construction

A curated taxonomically annotated database of anellovirus *orf1* gene sequences was created for genus and species identifications (see below).

First, a database of complete *orf1* sequences of the 161 ICTV anellovirus exemplars (VMR_MSL39_v1 [5]) from 33 genera was created, excluding gyroviruses, which are grouped together with other anelloviruses based on their genome organization similarity, but have very different genome and protein sequences. Then, a

collection of 30,921 anellovirus sequences was downloaded from the NCBI nt database (access date: 5 July 2024). To assign them into their respective genera, we extracted ORFs with a minimum length of 900 bases by using ORFfinder version 0.4.3 (https://ftp.ncbi.nlm.nih.gov/genomes/TOOLS/ORFfinder/), and queried them against the reference ICTV exemplars' *orf1* database by using BLASTN version 2.15.0 with the following options "-word_size 8 -reward 1 -penalty −1 -gapopen 4 -gapextend 1 -qcov_hsp_perc 50." A sequence was assigned to a specific genus if the top 10 BLASTN best hits with a minimum sequence identity of 60% were *orf1* sequences exclusively from that genus. The reference database was then supplemented with newly identified *orf1* sequences, and the remaining taxonomically unassigned ORFs were subsequently re-examined. The iterative search was performed until no additional sequences could be taxonomically assigned, producing a final collection of 19,325 *orf1* sequences across 33 anellovirus genera.

Next, within each genus, the *orf1* sequences were clustered together using the UCLUST algorithm implemented in USEARCH version 11.0.667 (27) with a sequence similarity threshold of 90%, resulting in 4,497 sequence clusters. Centroids were selected as the cluster representatives and were used to create the reference taxonomically annotated complete anellovirus *orf1* gene sequence database. We noted that 72 exemplars' sequences were not part of this database as they were not cluster centroids. These sequences were added back to the reference database.

## Anellovirus genus identification

We assigned the discovered anellovirus sequences into their respective genera based on the *orf1* sequence analysis. ORFs with at least 900 bases were extracted from the assembled sequences by using ORFfinder, and they were searched against the curated reference *orf1* database using BLASTN with the following parameters "-word_size 8 -reward 1 -penalty −1 -gapopen 4 -gapextend 1 -qcov_hsp_perc 50." A sequence was assigned to the genus of its BLASTN best hit if it showed 60% identity or greater. Assembled sequences lacking an *orf1* sequence of at least 900 bases were not included in the taxonomic analysis.

## Anellovirus species identification

The ICTV *Anelloviridae* Study Group recommends that a sequence be assigned to a species if its complete *orf1* sequence shows >69% similarity to at least one member of that species (4). Following this recommendation, for each genus including our newly discovered anellovirus sequences, we gathered complete *orf1* sequences, created sequence alignments for all pairs of sequences using MAFFT version 7.525 (28) with the parameters "--genafpair --maxiterate 1000," and computed their complete pairwise similarities. For each pair, lower- and upper-bound sequence similarities were calculated using equations 1 and 2, respectively:

$$S_\ell = \frac{N_\ell}{L_\ell}, \tag{1}$$

$$S_u = \frac{N_u}{L_u}, \tag{2}$$

where $S_\ell$ is the lower-bound of sequence similarity, $N_\ell$ is the total number of identical unambiguous base pairs, $L_\ell$ is the pairwise sequence alignment length, $S_u$ is the upper bound of sequence similarity, $N_u$ is the total number of positive base pairs (i.e., an ambiguous base is allowed to match with another ambiguous base or an unambiguous base), and $L_u$ is the alignment length, including only positions without gaps.

Next, an undirected graph of *orf1* sequences was created such that two sequences were connected when their sequence similarity exceeded 69% using igraph library version 1.6.0 (29) in R version 4.3.2 (30). This created two types of linkages: (i) "strict linkages"—sequences connecting through both the lower-bound and upper-bound

sequence similarity scores (i.e., showing both lower-bound and upper-bound similarity >69%), and (ii) "relaxed linkages"—sequences connecting only through the upper-bound sequence similarity scores (i.e., having the upper-bound similarity >69%, but not the lower-bound similarity).

A newly discovered anellovirus sequence was classified as belonging to a potentially novel species if it was a member of a cluster that, despite relaxed linkages, did not contain any ICTV exemplar or did not cluster with any other sequences. A sequence was classified as belonging to a known species if it was a member of a strict cluster that included only one ICTV exemplar sequence and was not part of a relaxed cluster containing multiple ICTV exemplars. A sequence was classified as having an undetermined species status for other conditions. Figure 1 illustrates some examples of this classification scheme.

## Similarity between anellovirus profiles of Thailand and other countries

To quantify the similarities between the anellovirus profile of Thailand and those of other countries, we employed pairwise probability Jaccard scores ($J_\mathcal{P}$) (31). For two probability distributions $X = (x_1, x_2, ..., x_n)$ and $Y = (y_1, y_2, ..., y_n)$, where $n$ is the dimension of the probability distributions, $x_i, y_i \in [0,1]$ for all $x_i, y_i$ and $\sum_i x_i = 1$, and $\sum_i y_i = 1$, their $J_\mathcal{P}$ score can be computed as follows:

$$J_\mathcal{P}(X, Y) = \sum_{x_i \neq 0, y_i \neq 0} \frac{1}{\sum_j \max\left(\frac{x_j}{x_i}, \frac{y_j}{y_i}\right)}. \qquad (3)$$

The country of origin for each sequence sourced from the NCBI nt database was extracted from the "geo_loc_name" field, and sequences lacking this information (524/21,295 = 2.46%) were excluded from the analysis. For anellovirus profile construction, relaxed clustering results were used, and non-cluster-representative sequences from the NCBI nt database ($n$ = 17,451) were re-assigned back to the relaxed clusters ($n$ = 368) of their representative sequences for the probability (i.e., frequency) calculation. $J_\mathcal{P}$ values were then computed between Thailand's profile and those of all other countries.

## Phylogenetic analysis

ORF1-protein phylogenetic analysis of our discovered anelloviruses was conducted, supplemented with translated sequences of our curated taxonomically annotated *orf1* reference sequences (see above), focusing on seven genera to which our viruses belong (*Alphatorquevirus*, *Betatorquevirus*, *Gammatorquevirus*, *Hetorquevirus*, *Lamedtorquevirus*, *Samektorquevirus*, and *Yodtorquevirus*). Sequences from each genus were individually aligned by using MAFFT with the parameters "--genafpair --maxiterate 1000." Multiple sequence alignments from different genera were subsequently aligned together using MUSCLE version 3.8.31 with the "-profile" option (32) to create a global sequence alignment. Then, the sequence alignment underwent manual curation to remove poorly aligned regions. Maximum-likelihood trees were reconstructed using IQ-TREE2 (33). The best-fit amino acid substitution model was automatically selected based on Bayesian Information Criterion scores by using ModelFinder (34) implemented in IQ-TREE2 (33). Clade support was computed using 10,000 bootstrap alignments with the ultrafast bootstrap approximation method (35). Individual genus trees were also created following the same protocol.

## RESULTS

### Anelloviruses are commonly found in Thai human sequencing data

This study analyzed non-human sequencing data obtained from 1,175 Thai human individuals. The reads were clean and assembled into a total of 8.35 million contigs, with a combined length of 13.42 gigabases. Read cleaning and assembly summary statistics are provided in Table S1.

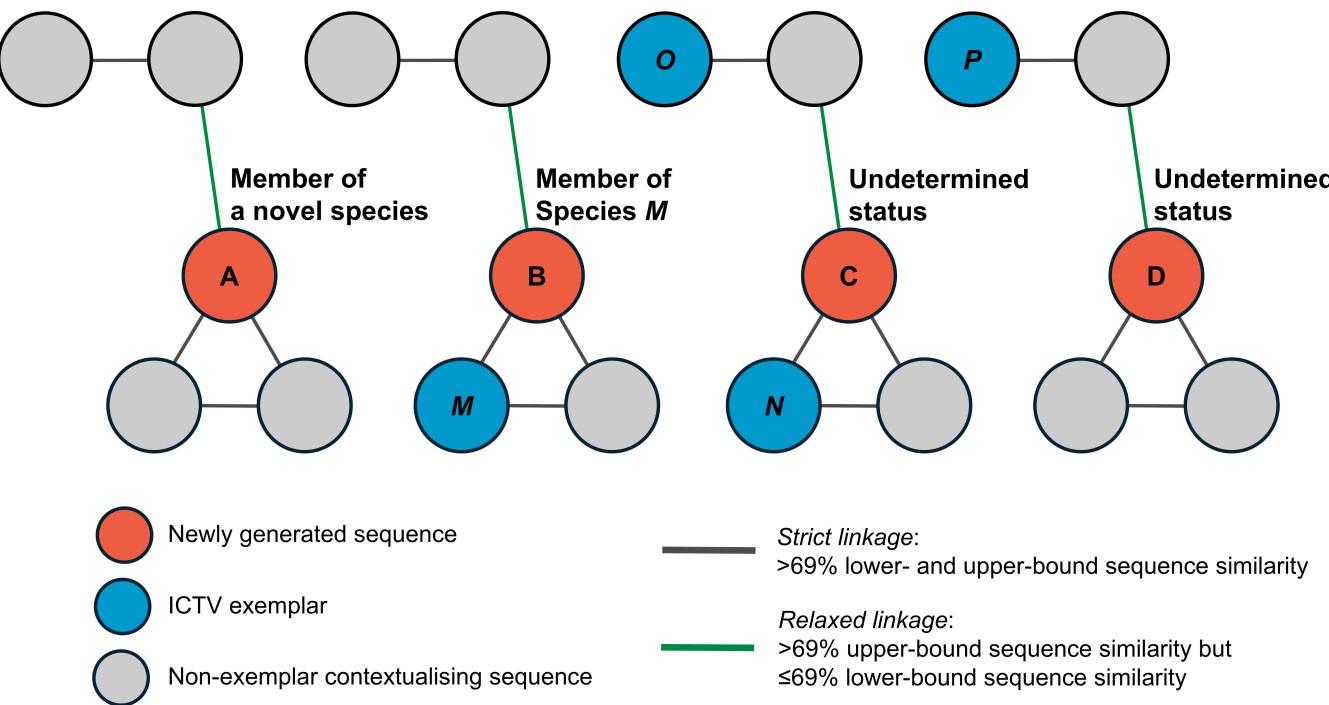

**FIG 1** Anellovirus species assignment based on pairwise sequence similarity network analysis. Sequences are represented by color-coded vertices. Gray edges denote "strict linkages," indicating that linked sequences show both lower- and upper-bound similarity >69%. Green edges denote "relaxed linkages," indicating that linked sequences show upper-bound similarity >69% but lower-bound similarity ≤69%. Species assignment is based on sequence clustering patterns formed using these two linkage types. In this hypothetical network, sequence A is classified as a member of a potentially novel anellovirus species because it belongs to a "relaxed cluster" lacking any ICTV exemplars (i.e., sequence A still does not show similarity above the threshold to any ICTV exemplars even when intermediate sequences are considered via relaxed linkages). Sequence B is classified to species M because it belongs to a "strict cluster" exclusively containing the ICTV exemplar of species M, and its relaxed cluster does not contain any additional ICTV exemplars (i.e., even only with strict linkages, sequence B still shows similarity above the threshold exclusively to the species M exemplar even when intermediate sequences are considered, and including relaxed linkages does not change that). Sequence C is assigned an undetermined status because it is part of a relaxed cluster with multiple ICTV exemplars (i.e., due to the possibility of belonging to a multi-species cluster, its species assignment hence remains unresolved here). For sequence D, while it is linked to one, and only one, ICTV exemplar, establishing this linkage requires at least one relaxed linkage, and they do not belong to the same strict cluster, so it is assigned an undetermined status (i.e., the assignment depends on linkage type used, and thus remains unresolved).

A search for anellovirus sequences within these contigs yielded 700 putative anellovirus sequences from 178 samples (15.15%). Among these, 108 contigs (15.43%) were annotated by CheckV as having "Complete" or "High-quality" genome quality (i.e., having a CheckV genome completeness of ≥90%) (Fig. 2). However, according to other additional criteria, it is important to note that not all of these were ultimately truly complete genomes; see below. In addition, among these 700 putative anellovirus sequences, we noted that 10 were very long, ranging from 12.67 to 121.97 kilobases and harboring 7–93 ORFs. These lengths are much longer than a typical anellovirus genome (2–4 kilobases), and none of the ORFs were identified as viral protein-encoding genes. These unusually long putative anellovirus sequences also received CheckV "Not-determined" ($n = 9$) and "Low-quality" ($n = 1$) quality scores. Altogether, these observations suggest that these sequences were likely misannotated or misassembled contigs.

Next, reciprocal BLAST analysis was performed. 622 sequences (622/700 = 88.86%) had anellovirus sequences from the NCBI databases as their BLAST best hits and were therefore deemed true positives (Fig. 2). The remaining 78 sequences did not return an anellovirus sequence as their best hit (11.14%), including all the unusually long sequences mentioned earlier. These sequences were thereby identified as false positives and were excluded from all subsequent analyses downstream. As a result, the number of anellovirus sequence-positive samples fell from 178 to 149, 12.68% of the total samples.

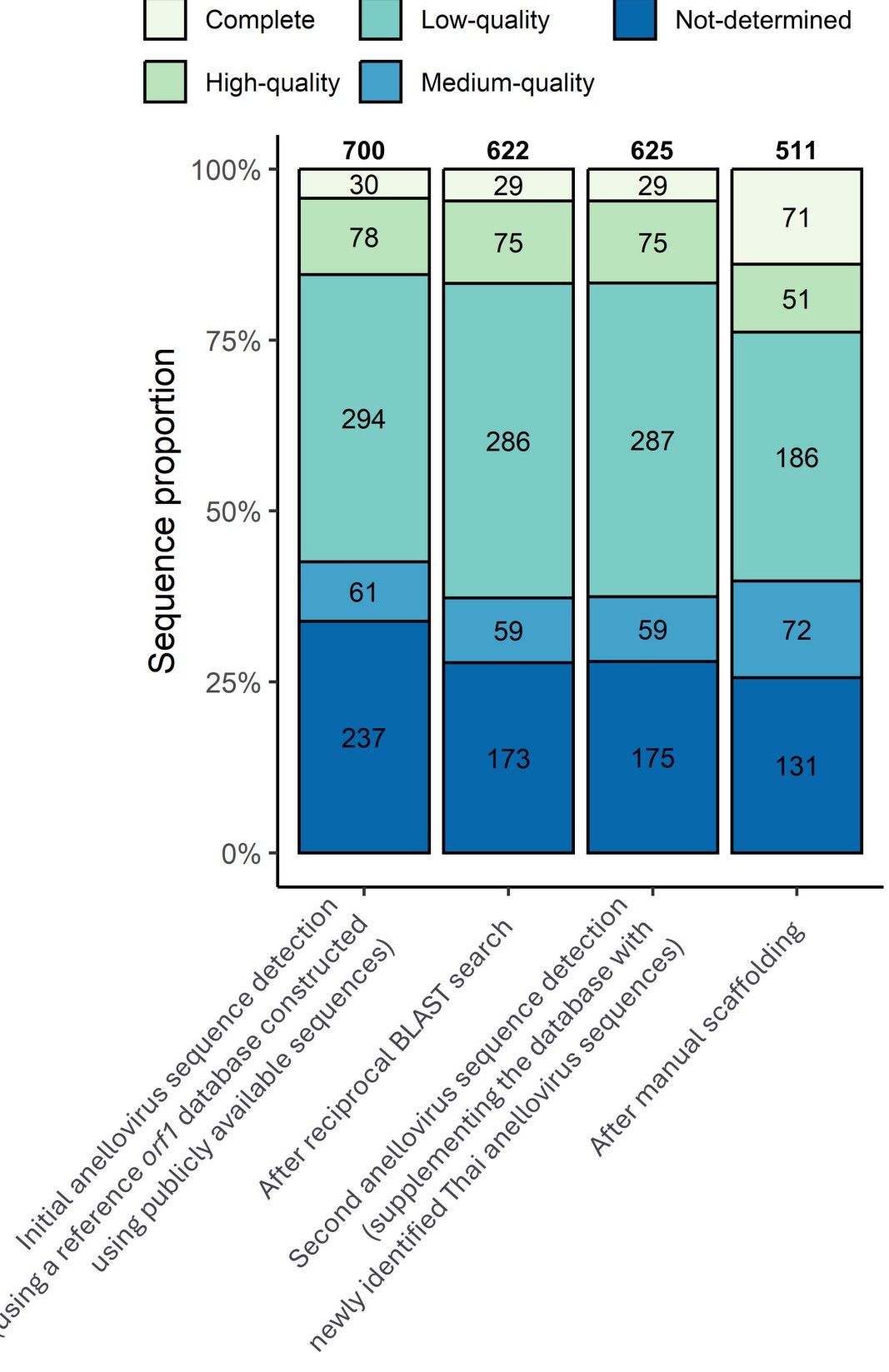

**FIG 2** Number of putative anellovirus sequences across detection stages. The assembly quality, as assessed by CheckV, is color-coded. Numbers above bars represent the total number of sequences, while numbers inside individual stacked bars represent the total number of sequences categorized into different assembly quality levels.

To detect anellovirus sequences that might be too different from the NCBI reference anellovirus sequences, predicted proteins from the true positives were added to the reference database, and the second iteration of sequence search was performed. The search found three additional putative anellovirus sequences of CheckV "Low-quality" (*n* = 1) and "Not-determined" (*n* = 2) assembly quality (Fig. 2). The reciprocal BLASTN was performed again, and their BLAST best hits were anellovirus sequences, corroborating the results. A third search iteration was performed with the reference database further supplemented with protein sequences from these three newly identified sequences, but no additional putative anellovirus sequences were detected.

To improve the genome assembly, clean reads were back-mapped to the contigs, and contig scaffolding was manually performed using paired-end information. This combined 207 contigs into 88 scaffolds, reducing the total anellovirus genome assemblies down to 511 (Fig. 2). This process substantially increased the number of CheckV "Complete" sequences and reduced the numbers of sequences with CheckV "Low-quality" and "Not-determined" quality (Fig. 2). Characterization of the 511 anellovirus genome assemblies revealed that 77 had the complete *orf1* gene sequence, identical sequences at both contig ends, and the conserved UTR upstream of the *orf1* gene, and, thus, they were deemed complete circular genomes of anelloviruses. The identical sequence on the 3′ end was removed, and the genome sequences were rotated such that the conserved motifs in the UTR were positioned at the 5′ end terminal (Fig. 3).

## Genus identification

The ICTV *Anelloviridae* Study Group established the taxonomy of anelloviruses at the genus level based on phylogenetic analysis of ORF1 protein sequences (4). Among the 511 anellovirus sequences discovered, 212 sequences (41.49%) were found to contain substantially long *orf1* sequences (≥900 bases), allowing them to be taxonomically assigned at the genus level using *orf1*. The remaining 299 sequences (58.51%), which either lacked *orf1* or had shorter *orf1* sequences, were excluded from taxonomic classification. To this end, the gene sequences were extracted and searched against the curated taxonomically annotated reference anellovirus *orf1* database by using BLASTN. Their taxonomic group was identified based on their BLASTN best hit.

Our sequences were identified as members of seven genera within the *Anelloviridae* family, including *Alphatorquevirus*, *Betatorquevirus*, *Gammatorquevirus*, *Hetorquevirus*, *Lamedtorquevirus*, *Samektorquevirus*, and *Yodtorquevirus* (Table 1). The majority of anelloviruses found in these sample sets were alphatorqueviruses (123 sequences, 58.02%) and betatorqueviruses (73 sequences, 34.43%) (see Table 1 for other genera). Seventy-seven were complete genomes belonging to four distinct genera: *Alphatorquevirus*, *Betatorquevirus*, *Gammatorquevirus*, and *Hetorquevirus* (Table 1), containing 2–7 ORFs with a minimum length of 300 bases.

Based on the genus of the closest ICTV exemplar, phylogenetic analysis of their ORF1 protein sequences produced results consistent with BLASTN-based analysis (Fig. 4). Examination of the phylogeny revealed that Thai anelloviruses do not form their own clades, and their diversity overlaps with that of published anelloviruses. This suggests that anelloviruses had entered Thailand multiple times from various regions and that cross-country transmission can readily occur.

## Pairwise sequence similarity network analyses to classify virus sequences into species

The ICTV *Anelloviridae* Study Group recommends that a sequence be assigned to a species if its complete *orf1* sequence shows >69% similarity to at least one member of that species (4). From the 511 putative anellovirus sequences, we were able to recover 134 complete *orf1* sequences from four genera, including *Alphatorquevirus* (82 sequences), *Betatorquevirus* (41 sequences), *Gammatorquevirus* (8 sequences), and *Hetorquevirus* (3 sequences). Following the ICTV *Anelloviridae* Study Group recommendation, we

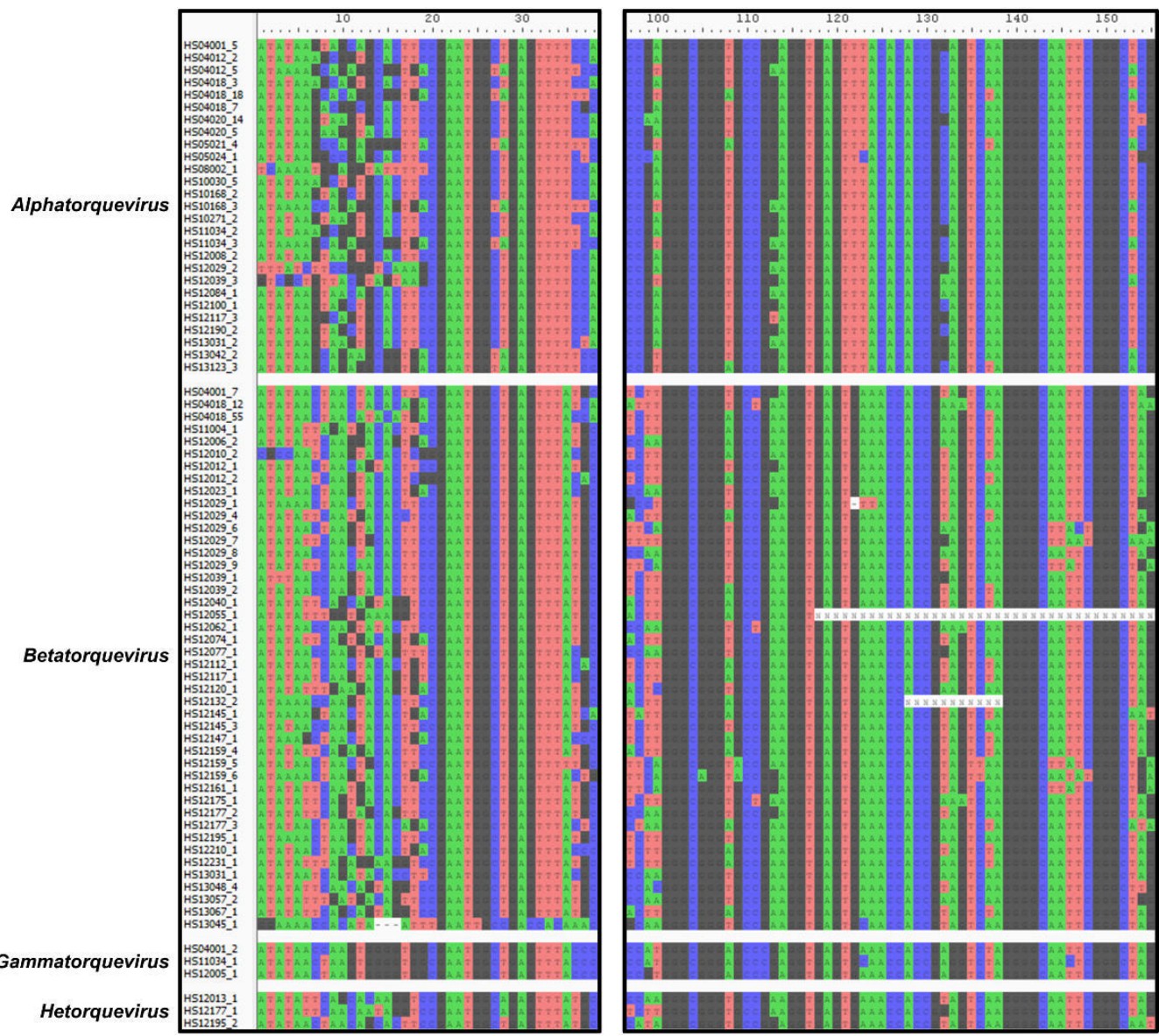

**FIG 3** Multiple sequence alignment of the complete anellovirus genomes showing four conserved motifs in the UTR upstream of the *orf1* gene. These include the TATA box (ATAWAW; positions 1–6), the conserved 15-base motif CGAATGGCTGAGTTT (positions 20–34), the Sp1 site GGGCGGG (positions 101–107), and the conserved 15-base motif AGGGGCAATTCGGGC (positions 138–152).

analyzed these complete *orf1* sequences to classify them at the species level by using pairwise sequence similarity network analysis with the similarity cutoff set at 69%.

The exact value of pairwise sequence similarity, however, depends on how the gaps and pairs of ambiguous bases are treated in the calculation. To this end, we therefore computed for each sequence pair upper- and lower-bound similarity values (for details, see Anellovirus species identification in Materials and Methods), and sequence pairs showing lower-bound similarities >69% were said to have "strict linkages," while those showing lower-bound similarities ≤69%, but upper-bound similarities >69% were said to have "relaxed linkages." Sequence clusters formed strictly by strict linkages are termed "strict clusters," and those formed by both strict and relaxed linkages are termed "relaxed clusters." The key idea was that if a sequence does not show similarity to any member of a known species above the 69% threshold level, even with relaxed linkages (i.e., being part of a relaxed cluster without any ICTV exemplar), then it is likely a member of a novel species. On the other hand, if a sequence still shows similarity above the 69% threshold

**TABLE 1** Number of Thai anellovirus sequences by genus[a]

| Genus | Number of sequences | Number of complete genomes |
|---|---|---|
| *Alphatorquevirus* | 123 | 27 |
| *Betatorquevirus* | 73 | 44 |
| *Gammatorquevirus* | 9 | 3 |
| *Hetorquevirus* | 3 | 3 |
| *Lamedtorquevirus* | 1 | 0 |
| *Samektorquevirus* | 2 | 0 |
| *Yodtorquevirus* | 1 | 0 |
| Total | 212 | 77 |

[a]Only sequences containing *orf1* were analyzed.

level to some members of a uniquely identifiable known species even with the lower-bound similarity (i.e., being part of a strict cluster containing exactly one ICTV exemplar) and not to any member of other species even with relaxed linkages considered (i.e., and that the relaxed cluster it is being a part of does not contain any other ICTV exemplar), then it is likely a member of that known species. Sequence clustering results are shown in Fig. 5 and 6, and detailed results can be found in Table S2.

Out of the examined 134 sequences, 32 sequences were found in 13 relaxed clusters containing only one ICTV exemplar, and they were also in strict clusters containing the ICTV exemplar (Fig. 5A through D). Thus, these 32 sequences were likely members of 13 known species (Fig. 6A). Fifty-five Thai anellovirus sequences were found to group (together with several non-Thai sequences) into 33 relaxed clusters lacking ICTV exemplars, including 6 alphatorquevirus clusters, 23 betatorquevirus clusters, and 4 gammatorquevirus clusters, indicating that they may be novel species (Fig. 6B). Interestingly, we found one cluster containing only two Thai betatorquevirus sequences and not sequences from any other country. These two sequences showed 99% pairwise sequence similarity to each other, and their most similar *orf1* sequence in the NCBI nt database (accession number: MN771720.1) showed only 64% similarity to them, supporting that they might represent a novel species of anelloviruses and highlight the unexplored and potentially unique anellovirus diversity in Thailand. The remaining 47 sequences were found in relaxed clusters with multiple ICTV exemplars or one ICTV exemplar, but they were in different strict clusters (Fig. 5A through D). Thereby, due to uncertainty, they were designated as having an "undetermined species" status (Fig. 6C).

Furthermore, we found 28 strict clusters to contain anellovirus sequences from multiple Thai individuals, suggesting that several species of anelloviruses have established their circulation within Thailand. In addition, among the 61 persons from whom complete *orf1* sequences could be retrieved, 24 (39.34%) were found to harbor anellovirus sequences from multiple strict clusters. Remarkably, one individual harbored sequences from 23 distinct strict clusters. These results demonstrate that a great diversity of anelloviruses can co-exist in one person, as had been previously reported (36–38).

## Thai anellovirus diversity in the global context

Based on the relaxed clustering networks (supplemented with 17,451 non-cluster-representative sequences from the NCBI nt database), we compared the anellovirus profile of Thailand against those of 31 other countries (n clusters = 368) by using the probability Jaccard Index $J_{\mathcal{P}}$ (31). We found that 22 countries across Africa (three countries), the Americas (three countries), Asia (seven countries), and Europe (nine countries) shared at least one relaxed cluster with Thailand, while the remaining nine countries did not (Fig. 7; Table S3). Switzerland showed the greatest similarity to Thailand ($J_{\mathcal{P}} = 0.4115$), followed by China ($J_{\mathcal{P}} = 0.3641$) and Spain ($J_{\mathcal{P}} = 0.3638$). Interestingly, despite their geographic proximity, Malaysia ($J_{\mathcal{P}} = 0.0411$) and Vietnam ($J_{\mathcal{P}} = 0.0217$) showed low similarity to Thailand (Fig. 7). These results support that Thailand does share anelloviruses with various countries; however, within this data set, the overall composition of the viruses appears distinct and does not necessarily correlate with geographic proximity.

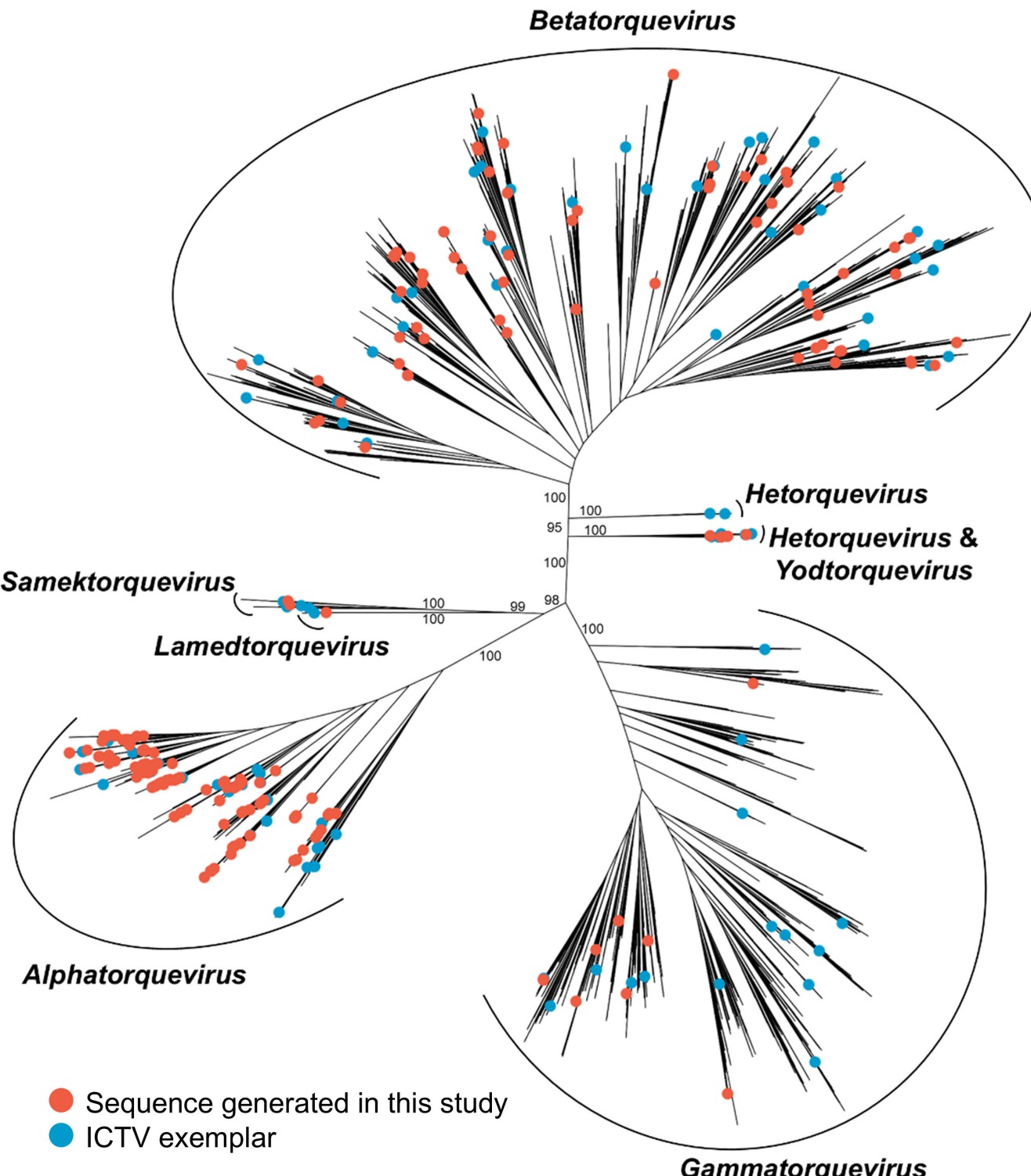

**FIG 4** Phylogenetic analysis of anelloviruses based on ORF1 protein sequences. The tree was reconstructed using a maximum likelihood method with the best-fit amino acid substitution model, LG+F+R10. Tips marked with orange and blue circles represent sequences generated in the study and those of the ICTV exemplars, respectively. Undecorated tips represent non-exemplar contextualizing sequences. Support values of major clades are shown, computed using 10,000 bootstrap alignments.

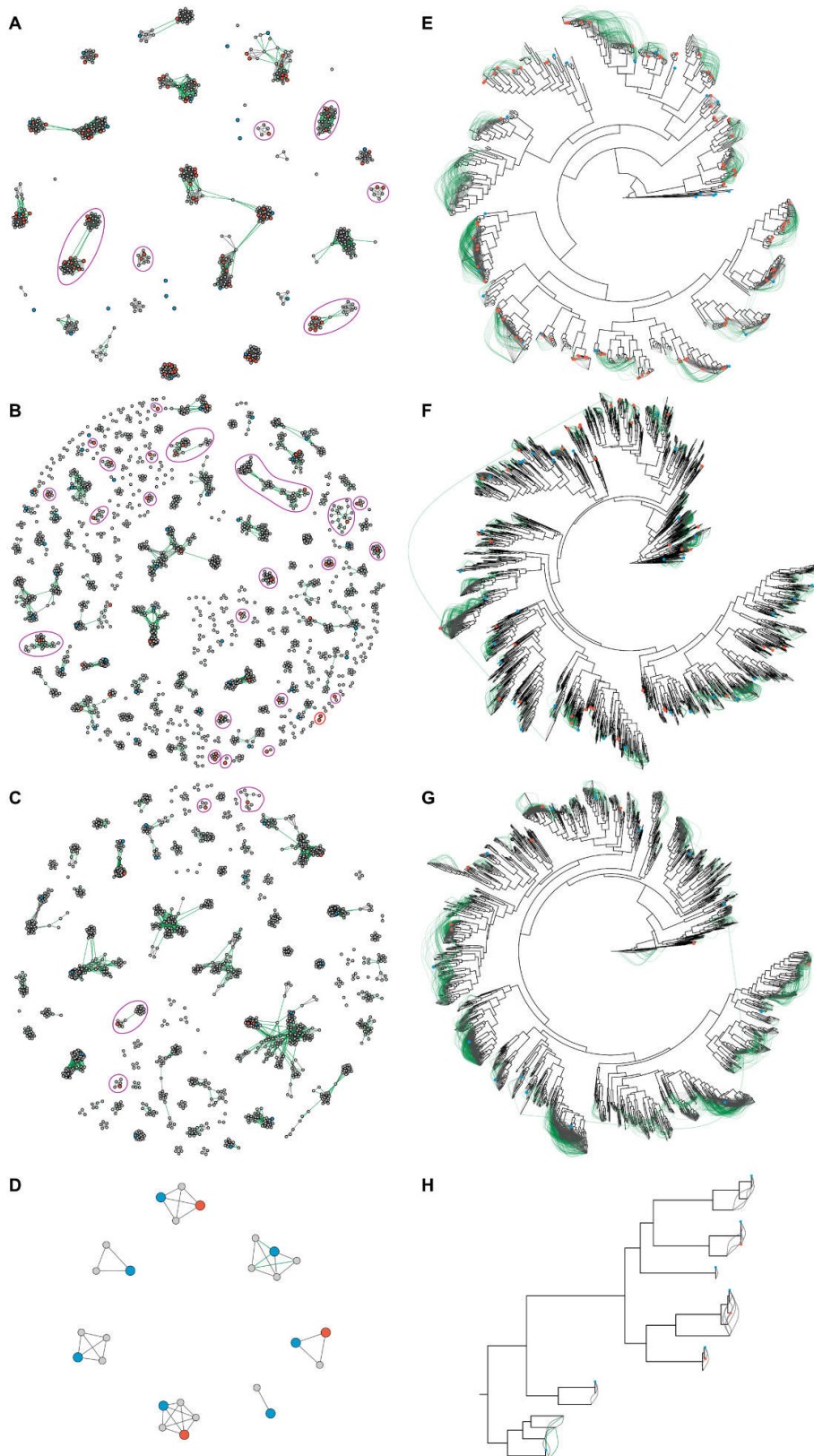

**FIG 5** *orf1* sequence similarity networks and phylogenies of alphatorqueviruses, betatorqueviruses, gammatorqueviruses, and hetorqueviruses. Panels A–D show *orf1* sequence similarity networks for alphatorqueviruses, betatorqueviruses, gammatorqueviruses, and hetorqueviruses, respectively. Orange, blue, and gray vertices represent complete *orf1* sequences

Fig 5 (Continued)

from the present study, from the ICTV exemplars, and from non-exemplar contextualizing sequences, respectively. Vertices representing sequences from this study and from ICTV exemplars are enlarged for visual emphasis. Gray edges denote "strict linkages," indicating that linked sequences show both lower- and upper-bound similarity >69%. Green edges denote "relaxed linkages," indicating that linked sequences show upper-bound similarity >69% but lower-bound similarity ≤69%. Clusters containing Thai anellovirus sequences without ICTV exemplars (i.e., potentially novel species) are outlined with purple lines, while clusters exclusively comprising Thai anellovirus sequences are outlined with red lines. Panels E–H show phylogenetic trees of ORF1 protein sequences for alphatorqueviruses (reconstructed using the LG+F+R8 model), betatorqueviruses (LG+F+R10), gammatorqueviruses (LG+F+R10), and hetorqueviruses (LG+F+Γ4), respectively. Green and gray curved lines denote relaxed and strict linkages, respectively. Trees are rooted using ICTV mutorquevirus exemplar sequences (not shown). Orange and blue solid circles indicate sequences from this study and ICTV exemplars, respectively; undecorated tips represent non-exemplar contextualizing sequences.

## Potential problems with pairwise sequence similarity analysis to classify anellovirus species and conflicts with phylogenetic analysis

We noted that one strict and five relaxed clusters in the genera *Alphatorquevirus*, *Betatorquevirus*, and *Gammatorquevirus* contained two ICTV exemplars (but not in the genus *Hetorquevirus*) (Fig. 5A through D). Table 2 shows an exhaustive list of all ICTV exemplars (and thereby current distinct species) that are in the same *orf1* sequence clusters. We found that, although they themselves showed both lower- and upper-bound sequence similarity <69%, they were linked together through some intermediate sequences sharing *orf1* sequence similarity above the species demarcation threshold of 69%. These results highlight a potential problem with the robustness of the pairwise sequence similarity-based classification.

Furthermore, given that the ICTV is now shifting its (official) virus taxonomy to an evolutionary-based taxonomy (39), we also examined how species assignments derived based on sequence similarity network analysis fit with the history of virus evolution (Fig. 5F through H). We found that, while there was a very strong correlation between the two, this was not without exceptions. Except for the genus *Hetorquevirus*, phylogenetic analysis revealed that sequences of the same strict and relaxed cluster could sometimes be polyphyletic. Furthermore, we found that there was one betatorquevirus cluster and one gammatorquevirus cluster that appeared to contain phylogenetically distantly related sequences linking through relaxed lineages (Fig. 5F and G). These results show that sequence similarity network analysis might not always give species classification results that are consistent with the virus's evolutionary history.

Regarding this issue of relaxed linkages linking distantly related sequences together, we suspected that this could be because of gaps in the sequence alignments, as had been previously noted (4). Indeed, the alignment of these relaxed pairs contained high average gap proportions (betatorquevirus pairs: 6.42%; gammatorquevirus pairs: 7.70% gap columns). To further investigate this, we computed gap proportions of all sequence pairs in these two betatorquevirus and gammatorquevirus relaxed clusters but in different strict clusters and compared them to those of sequence pairs in the same strict clusters. For pairs of sequences in the same relaxed cluster but in different strict clusters, the average gap proportion was estimated to be 7.75%, much greater than that of sequences within the same strict clusters, which was about 2.21%. We also found that the average difference between lower- and upper-bound similarities for pairs in different strict clusters was 4.48%, while pairs in the same strict clusters had a lower difference average of 1.58%, suggesting an association between gaps and similarity inflation.

Based on the definitions of the upper- and lower-bound similarity measurements provided in equations 1 and 2, the difference between the two bounds is proportional to the odds of a position in the pairwise alignment being a gap, and the bound difference becomes increasingly more and more pronounced with gap addition (Box 1). This means that a pair of sequences, despite having low sequence similarity, can appear to have high sequence similarity when their alignment contains a high number of gaps but is

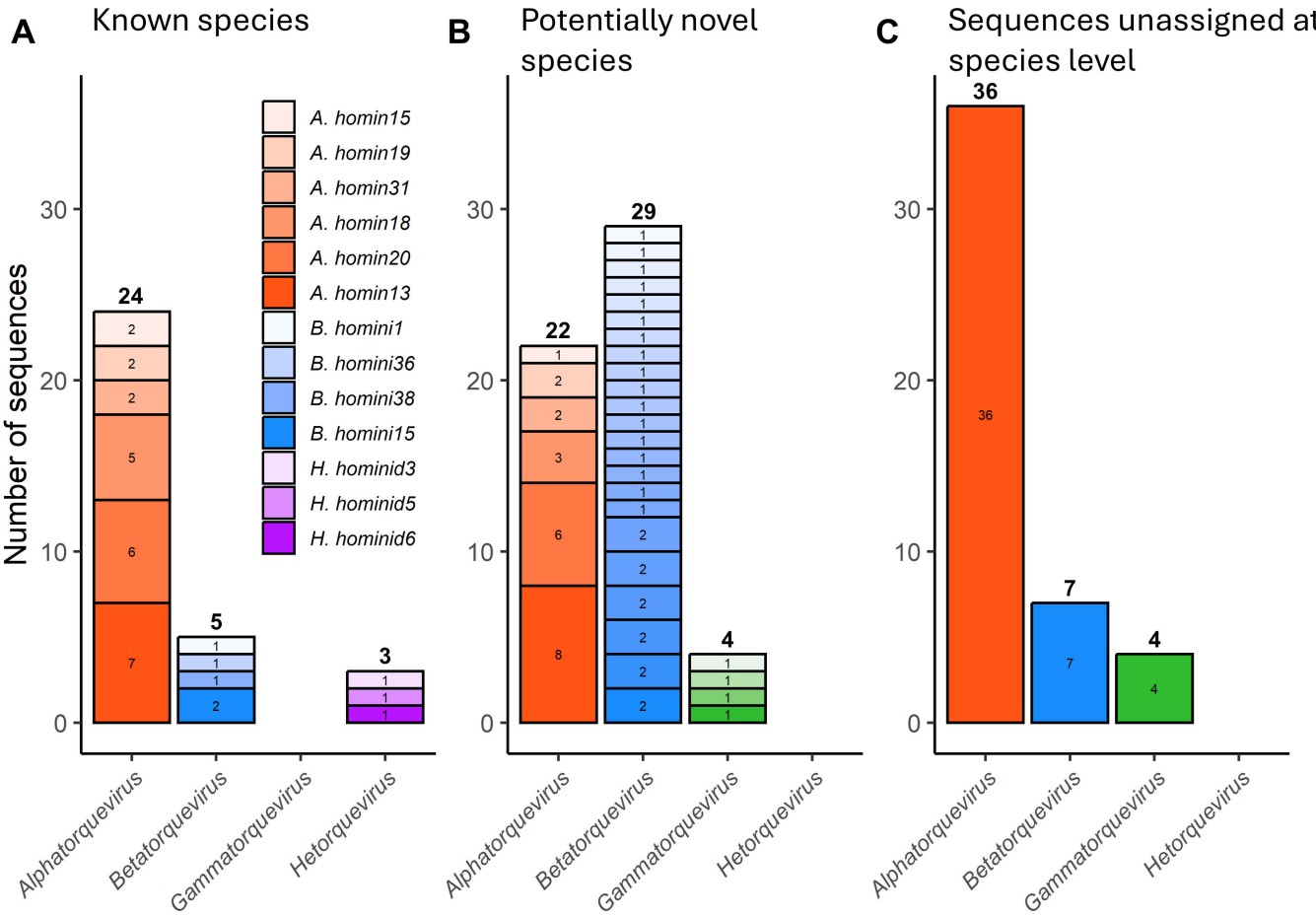

**FIG 6** Species assignments of Thai anellovirus sequences discovered in this study. Classification of Thai anellovirus sequences generated in this study was conducted based on complete *orf1* sequence similarity network analyses (see Fig. 5), using the classification scheme described in Fig. 1. Panels show the number of sequences of (A) known species, (B) potentially novel species, and (C) having undetermined species status. Each stacked bar in panels A and B represents an individual virus species/cluster.

excluded from the calculation, explaining the observed false classification of distantly related anellovirus sequences as the same species through relaxed linkages.

## DISCUSSION

Here, we screened non-human reads from 1,175 whole-genome sequencing data sets generated from Thai individuals for anellovirus sequences and report the most comprehensive collection of anellovirus sequences from Thailand. This includes 434 partial genome sequences and the first 77 complete genomes characterized from the Thai population. We found that 12.68% of the data sets detected positive for anellovirus sequences, falling within the previously reported range of 6% to 57% in Thai human samples (8, 9, 12). This detection rate is also comparable to the one recently reported from a study analyzing Japanese human whole-genome sequencing data, estimated at 12.3% (40).

The current classification scheme of anelloviruses was established based on *orf1* sequence analysis (4). Following this practice, this left our 299 sequences unclassified at least at the genus level (299/511 sequences = 58.51%) as they either contained very short partial *orf1* sequences or lacked the gene entirely. This large number of fragmented genome sequences was perhaps expected, given that we were limited only to short-read sequence data. To mitigate this potential issue, future metagenomic studies focusing on viral sequence discovery are therefore recommended to incorporate

**BOX 1. BOUND DIFFERENCE INCREASES WITH GAPS IN A PAIRWISE ALIGNMENT.**

Exclusion of gaps in a pairwise sequence alignment could artificially inflate sequence similarity. The difference between the upper- ($S_u$) and lower-bound ($S_\ell$) sequence similarities ($D$), assuming no ambiguous bases (i.e., $N_u = N_\ell$), can be computed as follows:

$$
\begin{aligned}
D &= S_u - S_\ell \\
&= \frac{N_u}{L_u} - \frac{N_\ell}{L_\ell} \\
&= \frac{N_\ell}{L_\ell - L_\ell G} - \frac{N_\ell}{L_\ell} \\
&= \frac{N_\ell L_\ell - N_\ell (L_\ell - L_\ell G)}{L_\ell (L_\ell - L_\ell G)} \\
&= \left(\frac{N_\ell}{L_\ell}\right)\left(\frac{G}{1 - G}\right) \\
&= S_\ell\left(\frac{G}{1 - G}\right),
\end{aligned}
\tag{4}
$$

where G is the proportion of gaps in the pairwise alignment and G/(1 − G) is the odds of a position in the pairwise alignment being a gap.

The rate of change of the bound difference with respect to the proportion of gaps ($dD/dG$) can be described as

$$
\frac{dD}{dG} = \frac{S_\ell}{(1 - G)^2}.
\tag{5}
$$

Equation 5 indicates that the bound difference becomes increasingly more and more pronounced with the addition of gaps. For instance, if we consider $S_\ell$ at 0.8 (indicating a pairwise similarity of 80%), the value of $dD/dG$ accelerates from 0.82 to 0.89 and then to 0.99 as $G$ increases from 0.01 (representing a gap proportion of 1%) to 0.05 (5%) and finally to 0.1 (10%).

long-read sequencing technologies (e.g., Oxford Nanopore, PacBio) alongside short-read methods to improve genome reconstruction and ORF validation.

Despite this limitation, however, our analysis still detected anellovirus sequences from seven genera, including *Alphatorquevirus*, *Betatorquevirus*, *Gammatorquevirus*, *Hetorquevirus*, *Lamedtorquevirus*, *Samektorquevirus*, and *Yodtorquevirus*. To our knowledge, *Alphatorquevirus*, *Betatorquevirus*, and *Gammatorquevirus* were the only genera of *Anelloviridae* that had ever been reported in the Thai human population (Table 3) (6–11, 14), and this study is the first to report *Hetorquevirus*, *Lamedtorquevirus*, *Samektorquevirus*, and *Yodtorquevirus* in the Thai humans. These results greatly expand our knowledge of anellovirus diversity in Thailand and could serve as a foundational data set for future anellovirus studies.

In addition, similar to another study (40), our results demonstrate that anellovirus sequences can be unexpectedly found in human sequencing data, even when such data sets are not specifically generated for virus discovery. This highlights the potential of human sequencing data as a valuable resource for uncovering unseen anellovirus diversity. However, data sets not originally designed for virus discovery, such as ones analyzed here, may lack appropriate negative/positive controls to detect viral sequence contaminants or to distinguish genuine viral sequences from non-viral sequences that exhibit detectable similarity to known viral genomes. It is therefore recommended that studies aiming to test specific hypotheses, such as potential associations between anelloviruses and clinical, biological, or environmental factors, should carefully validate

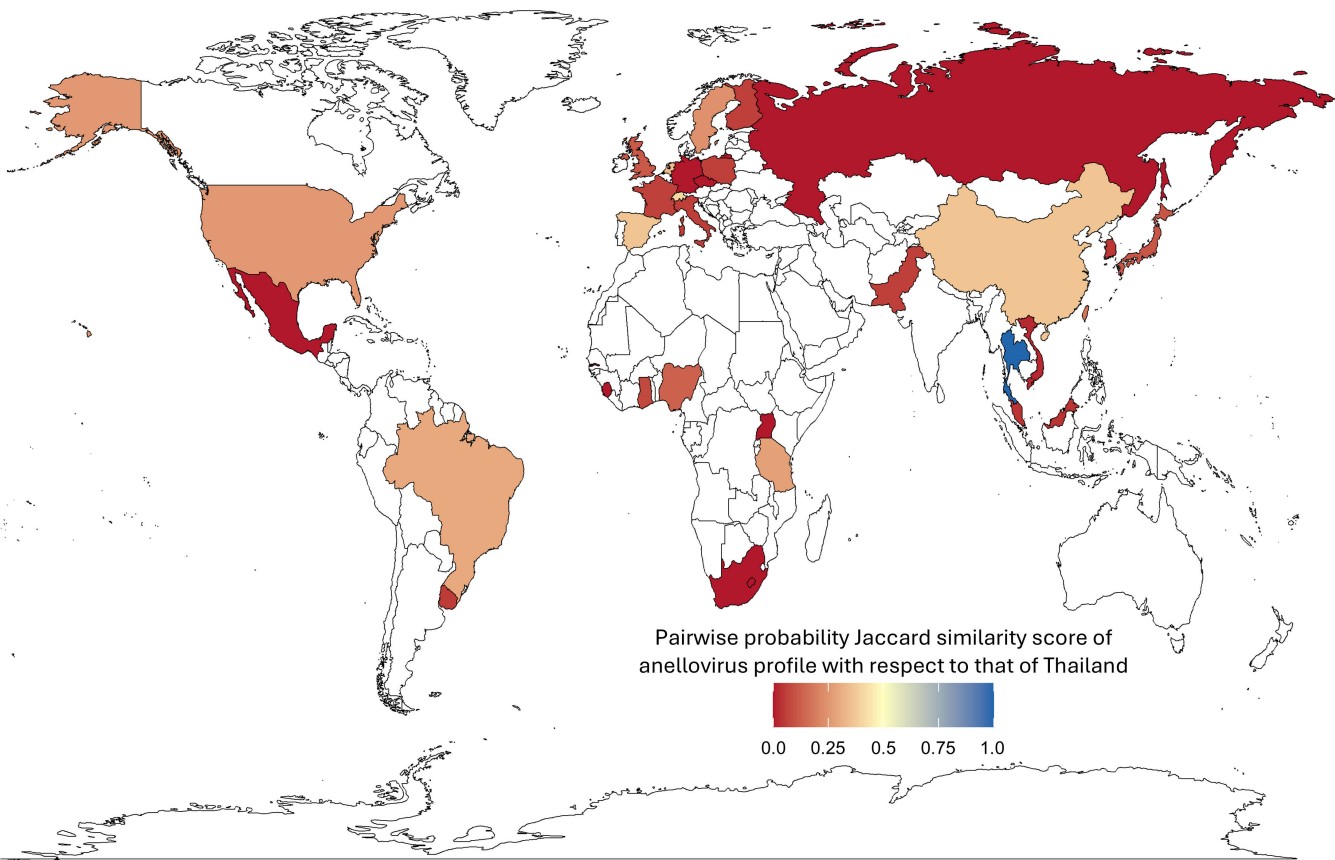

**FIG 7** Pairwise probability Jaccard score ($J_{\mathcal{P}}$) between anellovirus profiles of Thailand and each of the other countries. The map shows $J_{\mathcal{P}}$ values, comparing the anellovirus profile of Thailand with those of other countries derived based on relaxed sequence similarity clusters supplemented with 17,451 non-cluster-representative sequences from the NCBI nt database. Color scale indicates $J_{\mathcal{P}}$ values, ranging from 1 (high similarity; blue) to 0 (no shared clusters; red). Countries shown in white are those without anellovirus sequence data in our data set. The map was generated in R using the ggplot2 package (MIT license) and map data from the maps package (GPL-2 license).

sequence data and rigorously assess and interpret results, particularly when using viral sequences derived from data sets not originally intended for virome studies. That said, our analysis did not detect identical anellovirus sequences across multiple data sets, which suggests that our sequences are unlikely to be the result of systematic laboratory reagent contamination. Furthermore, reciprocal BLASTN searches consistently showed

**TABLE 2** Sequence clusters containing multiple species of anelloviruses

| Cluster type | Genus | Species | Pairwise sequence similarity (%) | |
| --- | --- | --- | --- | --- |
| | | | Lower-bound | Upper-bound |
| Strict | *Betatorquevirus* | *Betatorquevirus homini12* | 60.92 | 62.47 |
| | | *Betatorquevirus homini26* | | |
| Relaxed | *Alphatorquevirus* | *Alphatorquevirus homin1* | 60.98 | 65.63 |
| | | *Alphatorquevirus homin3* | | |
| | | *Alphatorquevirus homin21* | 58.18 | 62.11 |
| | | *Alphatorquevirus homin24* | | |
| | *Betatorquevirus* | *Betatorquevirus homini9* | 59.59 | 63.36 |
| | | *Betatorquevirus homini27* | | |
| | *Gammatorquevirus* | *Hetorquevirus hominid4* | 50.64 | 58.00 |
| | | *Hetorquevirus hominid15* | | |
| | | *Hetorquevirus hominid7* | 59.49 | 63.26 |
| | | *Hetorquevirus hominid14* | | |

TABLE 3  Anelloviruses reported in humans (*), gibbons (†), and pigs (‡) from Thailand[a]

| Genus | Source | | | | | | | |
|---|---|---|---|---|---|---|---|---|
| | This study* | (6)* | (7)* | (8)* | (9)* | (14)* | (10)† | (11)‡ |
| *Alphatorquevirus* | ✔ | ✔ | ✔ | ✔ | ✔ | ✔ | ✔ | |
| *Betatorquevirus* | ✔ | | | | | ✔ | | |
| *Gammatorquevirus* | ✔ | | | | | ✔ | | |
| *Hetorquevirus* | ✔ | | | | | | | |
| *Iotatorquevirus* | | | | | | | | ✔ |
| *Lamedtorquevirus* | ✔ | | | | | | | |
| *Samektorquevirus* | ✔ | | | | | | | |
| *Yodtorquevirus* | ✔ | | | | | | | |

[a]This table includes only studies that report the virus at the genus level or below.

that our sequences exhibited the highest similarity to known anelloviruses rather than to human (or other cellular) sequences, suggesting that they are unlikely to be misidentified host-derived fragments. Taken together, these support the validity of our newly identified anellovirus sequences and suggest that they may be suitable for use in future studies, at least those focusing on the investigation of anellovirus diversity.

Recent studies analyzing HTS data have reported multiple distinct lineages of anelloviruses co-existing within single individuals (36–38, 41). In line with these findings, we found 24 out of 61 samples in our data set from which complete *orf1* sequences could be recovered to contain multiple distinct anellovirus sequences (i.e., belonging to different strict clusters). These could be a result of sequential infections, or a single "bulk transmission" into the persons, or a combination of both. Indeed, a recent study showed that blood transfusion can introduce multiple lineages of anelloviruses to recipients (36). Given that a notable proportion of our data (24/61 = 39.34%) appeared to contain multiple lineages of anelloviruses, it may be of interest to explore the potential immunological or clinical implications of such co-infections. Although several previous studies have already linked overall anellovirus load with host immune status, cancer, and the severity of several diseases in humans (3, 40, 42–44), anelloviruses are often treated as a single viral population, and the effects of co-infecting lineages have remained understudied. Further functional and systemic investigations are needed to shed more light on the roles of anellovirus diversity and co-infection in human health and disease.

Furthermore, our analysis revealed that Thai anellovirus sequences were distributed among clusters that also contained sequences from multiple countries across different geographical regions (Fig. 7), and they did not form a single clade (Fig. 4). Nevertheless, within our data set, the anellovirus profile of Thailand was quite distinct from those of other countries (e.g., $J_{\mathcal{P}} < 0.5$ for all countries), and interestingly, the Thai profile appeared more similar to those of European countries and China than to those of neighboring countries (Fig. 7). These results suggest that the virus had entered the Thai human population multiple times from diverse sources, and international cross-transmission can readily occur. Further study is required to examine in more detail how anelloviruses spread in and out of Thailand as well as within the country.

With sequence similarity network analysis, we identified 55 of our anellovirus sequences as belonging to 33 potentially new species, 32 sequences belonging to 13 known species, and 47 sequences with undetermined status. The new species were in the genera *Alphatorquevirus* (six potentially novel species), *Betatorquevirus* (23 potentially novel species), and *Gammatorquevirus* (four potentially novel species). Although a recent study predicted that the diversity of these genera is nearing saturation (45), our results suggest that their diversity may not yet be fully explored. This is particularly evident for the genus *Betatorquevirus*, for which our analysis identified 23 potentially new species (69.70% of all new species detected) and, intriguingly, with one cluster containing solely sequences from Thailand. These findings illustrate a considerable and largely unexplored anellovirus diversity in Thailand, and at the same time highlight the potential of human sequencing data as a useful resource for future virome research and virus discovery.

Lastly, our analyses highlight some challenges and limitations of applying a pairwise similarity-based classification scheme to anelloviruses. One challenge is that the current classification scheme of anelloviruses relies on one key underpinning assumption that there exists a considerable gap in genetic diversity at which point a single threshold can be placed to robustly demarcate virus species (4, 46). From an evolutionary perspective, this is not completely unreasonable. As obligate parasites, the evolution and adaptation landscape of viruses is strongly constrained by their host environment (47, 48). The strong natural selection pressure that viruses constantly experience can cause rapid lineage turnover and extinction of *"unfit"* variants, which subsequently can create large gaps in virus genetic diversity, at least in theory. However, as previously shown (4), while there indeed exists a trough around ~69% pairwise *orf1* sequence identity, it is not completely empty (i.e., there exist pairs of viruses that show *orf1* sequence similarity around 68%–70%). This, in turn, suggests that the current classification scheme can be somewhat sensitive to the reference data set used, and to some extent unstable, particularly when it involves anelloviruses showing pairwise *orf1* sequence similarity values near the 69% threshold.

Here, we found that, with sufficient virus diversity included in the network analysis, it is possible for multiple ICTV species exemplars to be grouped together in the same sequence cluster linking through (potentially several) intermediate sequences even though they themselves share *orf1* sequence similarity <69%. While these exemplars are currently classified as distinct species, our sequence similarity network analysis indicated that, with sufficient sampling efforts, distinct species may be classified as belonging to the same species under the current classification framework. As we continue to investigate anellovirus diversity from more and more underexplored geographical regions and human populations, the observed gap in the virus genetic diversity will likely become less and less pronounced. In theory, it is therefore possible that, with continual sequencing efforts and novel anellovirus discoveries, a large and dense network of anellovirus sequences might emerge, unifying many species of anelloviruses together under the current pairwise similarity-based classification scheme. While this is entirely hypothetical and speculative, it highlights a possible theoretical limitation of the current classification scheme in the face of increasing sampling.

Furthermore, the ICTV recently published a consensus statement that virus taxonomy should now be based on evolutionary relatedness (39). However, the current anellovirus species classification is not based on their evolutionary history, but relies entirely on pairwise similarity analysis of viruses' complete *orf1* sequences (2). Indeed, pairwise sequence similarity analysis can capture phylogenetic relationships well in general, but sometimes the two analyses can give inconsistent results (49). In this study, our results showed that, while pairwise *orf1* gene sequence similarity analysis gave consistent results with phylogenetic analysis at the genus level, the two gave conflicting results in several cases at the species level. Some sequence clusters were not monophyletic, and this could be observed not only with relaxed clusters, but also with strict clusters, in which sequences were linked together through lower-bound sequence similarity already (Fig. 5A through C and E through G). Moreover, we also found that phylogenetically distantly related sequences could be classified as belonging to the same sequence clusters if gaps are ignored in pairwise sequence similarity calculation and if ambiguous bases are allowed to match with other ambiguous bases (or unambiguous bases) (Fig. 5F and G). Indeed, pairwise similarity can be significantly affected by how gaps in the alignments are treated (Box 1). Thus, relationships derived based on overall similarity should be carefully examined and cross-checked with phylogenetic analysis to ensure that the resultant classification is consistent with the evolutionary history of the viruses, especially when the similarity values are calculated from pairwise alignments with high gap proportions. In fact, given that the ICTV is now shifting toward an evolutionary-based taxonomy (39) and more novel anellovirus sequences are being discovered from underexplored regions and populations, the current classification scheme of *Anelloviridae* may therefore need to be revised. Now that HTS technologies

have become readily accessible around the globe, we can expect to see more and more novel anelloviruses being reported, and this will undoubtedly transform the landscape of anellovirus taxonomy.

## ACKNOWLEDGMENTS

The authors thank Bharkbhoom Jamsai for advice on visualization with R.

Conceptualization: P.A.; Methodology: P.A. and W.P.; Resources: C.N. and S.T.; Data curation: C.N., J.P., and W.P.; Investigation: P.A. and W.P.; Formal analysis: P.A. and W.P.; Visualization: P.A. and W.P.; Writing—original draft: P.A., S.T., and W.P.; Writing—review and editing: P.A., S.T., and W.P.; Supervision: S.T. and P.A.; Funding acquisition: P.A.

This study was supported by the National Science and Technology Development Agency, Thailand (JRA-CO-2563-12568-TH), and Mahidol University (MU's Strategic Research Fund): fiscal year 2023 [MU-SRF-WC-02B/66].

## AUTHOR AFFILIATIONS

[1]Department of Microbiology, Faculty of Science, Mahidol University, Bangkok, Thailand
[2]Pornchai Matangkasombut Center for Microbial Genomics, Department of Microbiology, Faculty of Science, Mahidol University, Bangkok, Thailand
[3]National Center for Genetic Engineering and Biotechnology, National Science and Technology Development Agency, Pathum Thani, Thailand

## AUTHOR ORCIDs

Worakorn Phumiphanjarphak ⓘ http://orcid.org/0000-0002-8879-5228
Pakorn Aiewsakun ⓘ http://orcid.org/0000-0002-5665-4041

## FUNDING

| Funder | Grant(s) | Author(s) |
|---|---|---|
| National Science and Technology Development Agency | JRA-CO-2563-12568-TH | Pakorn Aiewsakun |
| Mahidol University | MU-SRF-WC-02B/66 | Pakorn Aiewsakun |

## AUTHOR CONTRIBUTIONS

Worakorn Phumiphanjarphak, Data curation, Formal analysis, Investigation, Methodology, Visualization, Writing – original draft, Writing – review and editing | Jinjutha Parkbhorn, Data curation | Chumpol Ngamphiw, Data curation, Resources | Sissades Tongsima, Resources, Supervision, Writing – original draft, Writing – review and editing | Pakorn Aiewsakun, Conceptualization, Formal analysis, Funding acquisition, Investigation, Methodology, Supervision, Visualization, Writing – original draft, Writing – review and editing

## DATA AVAILABILITY

The anellovirus sequences generated in this work are publicly available in the European Nucleotide Archive with the accession numbers OZ258605 to OZ258816 and OZ285112 to OZ285410.

## ADDITIONAL FILES

The following material is available online.

Supplemental Material

**Supplemental material (Spectrum00866-25-s0001.xlsx).** Tables S1 to S3.

Open Peer Review

**PEER REVIEW HISTORY (review-history.pdf).** An accounting of the reviewer comments and feedback.

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
