## [Reviewer comments · Microbiology Spectrum]

Microbiology Spectrum

Discovery of diverse anellovirus sequences in Thai human sequencing data

Worakorn Phumiphajarphak, Jinjutha Parkbhorn, Chumpol Ngamphiw, Sissades Tongsimma, and Pakorn Aiewsakun

Corresponding Author(s): Pakorn Aiewsakun, Mahidol University Faculty of Science

Review Timeline:

Submission Date:	March 24, 2025
Editorial Decision:	April 20, 2025
Revision Received:	June 26, 2025
Editorial Decision:	July 13, 2025
Revision Received:	July 25, 2025
Accepted:	July 27, 2025

Editor: Biao He

Reviewer(s): Disclosure of reviewer identity is with reference to reviewer comments included in decision letter(s). The following individuals involved in review of your submission have agreed to reveal their identity: Chutchai Piewbang (Reviewer #2)

Transaction Report:

DOI: <https://doi.org/10.1128/spectrum.00866-25>

Re: Spectrum00866-25 (Discovery of diverse anellovirus sequences in Thai human sequencing data)

Dear Dr. Pakorn Aiewsakun:

Thank you for the privilege of reviewing your work. Below you will find my comments, instructions from the Spectrum editorial office, and the reviewer comments.

Revision Guidelines

Sincerely,
Biao He
Editor
Microbiology Spectrum

Reviewer #1 (Public repository details (Required)):

The sequences (de novo assembled) currently are only available as supplementary data associated with this paper. They have not been deposited in public databases. The authors can submit these, if they choose, as third part annotations to GenBank.

Reviewer #1 (Comments for the Author):

The MS titled "Discovery of diverse anellovirus sequences in Thai human sequencing data" address a significant gap in the knowledge of anelloviruses that circulating within the human population in Thailand. The work is based on data minging of

human sample whole genome sequence data that is publicly available. I like the approach the authors have taken to try address classification and yes, 69% PI is not an ideal way to address the species level taxonomy.

The MS is generally well written, and I have only a few minor comments. I do have some broader questions with regard to data analysis, especially de novo assembled contigs assembled to genome or larger contigs by terminal matching and sequence data.

1) I recommend the use of the full names when referring to viruses - alpha-, beta- etc please change to alphatorqueviruses, betatorqueviruses etc. through out the MS.

2) We have moved past the term next generation sequencing, we are already in 3rd / 4th generation of sequencing. Rather use high through put sequencing.

3) Line 99, 125 and line 215-217: I would like to highlight that CheckV may not be the best toll to determine completeness of small viruses with overlapping ORFs. The better way to check for completeness for small circular DNA viruses is terminal redundancy in de novo assembled contigs.

4) Line121: With the manual scaffolding are you worried about generating in silico chimeras? Various anelloviruses have shown that these viruses are highly recombinant with recombination hot spots in the non-coding regions.

5) All the partial and complete genomes are only available in the supplementary data as a fasta file. Why not deposit these in GenBank using the third part annotations set up. As it stands none of these sequences will be database searchable and with no GenBank accession numbers, they will not get classified.

Reviewer #2 (Public repository details (Required)):

Authors analyse the genome sequences of anelloviruses and some of the detected viruses are novel strains

Reviewer #2 (Comments for the Author):

The manuscript investigates the diversity of anelloviruses in Thai individuals by analyzing 1,175 whole genome sequencing (WGS) datasets. The authors identified 511 anellovirus genomes, including 77 complete and 434 partial genomes, spanning seven genera. Notably, four of these genera-Hetorquevirus, Lamedtorquevirus, Samektorquevirus, and Yodtorquevirus-have not previously been reported in Thailand. Through ORF1-based taxonomic and phylogenetic analyses, the authors propose 33 potentially novel species and discuss inconsistencies in the current ICTV classification system, particularly due to the influence of alignment gaps on pairwise similarity measures.

The manuscript presents high-quality, timely research with substantial potential impact. However, several claims-particularly those related to species classification, evolutionary interpretation, and the degree of novelty-require more cautious framing and further clarification. I recommend major revisions focused on methodological clarification, improved comparative context, and more balanced discussion of results and limitations.

Line 24-26: The claim of 77 complete and 434 partial genomes from 149 samples requires clearer explanation of the criteria used to define "completeness." Was completeness determined solely by CheckV, or were other factors such as circularity or presence of conserved genomic motifs also considered?

Lines 85-87: The manuscript does not mention the inclusion of negative controls or contamination assessment. Given that the WGS data were originally generated for human genomic studies, the possibility of detecting false positives or host-derived sequences mimicking viral content must be addressed.

Lines 87-95 and 212-217: The presence of very long contigs (12-121 kb), far exceeding typical anellovirus genome sizes (~2-4 kb), raises concerns about misassembly or chimeric artifacts. Although the manuscript notes these are likely misassembled, it is unclear how many were ultimately excluded from the analysis and based on what threshold. Please clearly state the filtering strategy for removing implausible contigs.

Lines 128-151 and 258-261: Genus assignment is limited to sequences with ORF1 {greater than or equal to}900 bp, which leaves 299 of the 511 genomes unclassified. The authors should clarify whether any supplementary classification methods were considered for these partial genomes, such as using other ORFs or UTR motifs, or if they were entirely excluded from taxonomic analysis.

Line 217-221: Please specify whether the anomalously long sequences mentioned were completely excluded from downstream analysis. The current language leaves ambiguity regarding whether some might have been retained or mistakenly used.

Lines 407-409: While the limitations of short-read sequencing are briefly acknowledged, the discussion would benefit from a more explicit recommendation for the use of long-read sequencing technologies (e.g., Oxford Nanopore, PacBio) in future studies to improve genome reconstruction and ORF validation.

Line 412-414: The claim that Heterotorquevirus, Lamedtorquevirus, Samektorquevirus, and Yodtorquevirus are newly reported in Thailand is supported. However, it would significantly strengthen the impact of this finding to include a summary comparison table of previously reported genera in Thailand (based on references 6-15) versus those identified in this study.

Line 416-419: No virome controls, spike-ins, or mock samples are mentioned. A brief discussion on the potential for contamination, especially since the datasets were not intended for virome studies, is necessary to strengthen the reliability of the viral detection claims.

Line 419: The use of the phrase "treasure trove" is non-academic and should be replaced with more neutral language such as "valuable dataset" or "rich source."

Line 423-426: The brief mention of co-infection with multiple anelloviruses should be expanded. The authors are encouraged to include a discussion of the potential immunological or clinical implications of co-infection, referencing recent literature on host immune modulation.

Line 440-441: The statement that these findings "warrant further systematic screening for novel anellovirus discovery" is too strong in the absence of any functional validation. Please consider rephrasing this to suggest that the results "support the potential utility" of WGS data for future virome research.

Lines 449-454: The claim that the classification scheme is unstable due to reference dataset dependency is overstated. While the study shows variability around the 69% similarity threshold, this does not in itself prove instability across the classification system. A more measured phrasing is recommended.

Line 454-460: The idea that increased sampling could collapse distinct species into a single mega-cluster under the current classification criteria is an interesting hypothesis but remains speculative. Please qualify this as a theoretical concern rather than a demonstrated outcome.

Line 483-485: The suggestion that all classification results should be cross-validated with phylogenetic analysis is valid. However, the phrasing implies a frequent conflict between methods. Please soften this claim unless the manuscript quantifies the frequency and impact of such discrepancies.

Figures 5 & 6: These figures are important for interpreting species boundaries and novel clusters. However, the legends should be expanded to define what "strict" and "relaxed" clusters represent and how these classifications were applied in the study. For readers unfamiliar with similarity network analysis, this added context would improve accessibility.

Table S2: While valuable, this supplementary table would be more informative if key data-such as the number of sequences assigned to known species, novel species, or with undetermined status-were summarized within the main text.

Additional Suggestion: The manuscript would benefit from a visual or tabular summary of newly identified species by genus and geographic origin (if such metadata exist). This would help highlight the uniqueness of the Thai anellovirus dataset within the global context.

Furthermore, the importance section should be revised as it is not the summary of your findings or repeating the information within the abstract.

Point-by-point response to reviewer's comments

Reviewer #1:

The MS titled "Discovery of diverse anellovirus sequences in Thai human sequencing data" address a significant gap in the knowledge of anelloviruses that circulating within the human population in Thailand. The work is based on data mining of human sample whole genome sequence data that is publicly available. I like the approach the authors have taken to try address classification and yes, 69% PI is not an ideal way to address the species level taxonomy.

The MS is generally well written, and I have only a few minor comments. I do have some broader questions with regard to data analysis, especially de novo assembled contigs assembled to genome or larger contigs by terminal matching and sequence data.

Reply: We thank the reviewer for evaluating our work and positive comments. Specific points are addressed below.

Comment 1.1: 1) I recommend the use of the full names when referring to viruses - alpha-, beta- etc please change to alphatorqueviruses, betatorqueviruses etc. through out the MS.

Reply 1.1: We thank the reviewer for their suggestion, and as recommended, we now use virus full names in our revised manuscript.

Line 414-418: To further investigate this, we computed gap proportions of all sequence pairs of sequences in these two betatorqueviruses ~~beta-~~ and gammatorquevirus relaxed clusters but in different strict clusters, and compared to those of sequence pairs in the same strict clusters, ~~but which did not show this problem.~~

Comment 1.2: 2) We have moved past the term next generation sequencing, we are already in 3rd / 4th generation of sequencing. Rather use high though put sequencing.

Reply 1.2: We thank the reviewer for their suggestion, and as recommended, we now use the term "high throughput sequencing (HTS)" instead of "next-generation sequencing (NGS)" throughout the manuscript.

Line 84-86: With high throughput sequencing (HTS) ~~next-generation sequencing (NGS)~~ technologies, a number of studies from Thailand recently reported these viruses from humans (12–14) and macaques (15).

Line 89-90: In this study, we report the identification of anellovirus sequences mined from 1,175 HTS ~~NGS~~-datasets derived from Thai human individuals.

Line 100-104: All sequence ~~NGS~~-datasets analysed in this study were from an in-house database maintained by human read-subtracted datasets obtained from the National Center for Genetic Engineering and Biotechnology (BIOTEC), Thailand. All were paired-end short reads generated by HTS technologies with human reads removed through read-mapping against the reference human genome hg38.

Line 482-483: Recent studies analysing HTS NGS data have reported multiple distinct lineages of anelloviruses co-existing within single individuals (36–38, 41).

Line 573-576: Now that HTS NGS technologies have become readily accessible around the globe, we can expect to see more and more novel anelloviruses be reported, and this will undoubtedly transform the landscape of anellovirus taxonomy.

Comment 1.3: 3) Line 99, 125 and line 215-217: I would like to highlight that CheckV may not be the best tool to determine completeness of small viruses with overlapping ORFs. The better way to check for completeness for small circular DNA viruses is terminal redundancy in de novo assembled contigs.

Reply 1.3: We understand the reviewer's concern (which was also raised by **Reviewer #2; Comment 2.1**), and we appreciate their insightful recommendation. In this work, we actually only used CheckV as a preliminary tool to assess the quality of our sequence assemblies. To determine the genome completeness, we evaluated the completeness of the *orf1* gene, and looked for the presence of terminal redundancy, and the known conserved untranslated region (UTR) located upstream of the *orf1* gene.

We apologise that this was unclear in our original manuscript, and we have now revised the Abstract, Methods, and Results sections to better describe the criteria used to define genome completeness.

Line 26-30: Our analyses uncovered 77 complete and 434 partial anellovirus genomes from 149 samples (12.68%), detected anellovirus sequences in 149 datasets (12.68%), uncovering 434 partial anellovirus sequences, and 77 complete genome sequences, characterised by the presence of terminal redundancy, complete *orf1*, and the conserved untranslated region upstream of the *orf1* gene.

Line 163-166: Sequence quality was assessed again. Finally, sequence quality was assessed using CheckV version 1.0.1 (20) with the database version 1.5, and, finally, genome sequence completeness was determined based on the presence of terminal redundancy, complete *orf1*, and the conserved untranslated region (UTR) located upstream of the *orf1* gene.

Line 295-298: Characterisation of Among the 511 anellovirus genome assemblies revealed that, 77 of them had the complete *orf1* gene sequence, identical sequences at both contig ends, and the conserved UTR upstream of the *orf1* gene, and, thus, they were deemed to be complete circular genomes of anelloviruses.

Comment 1.4: 4) Line 121: With the manual scaffolding are you worried about generating in silico chimeras? Various anelloviruses have shown that these viruses are highly recombinant with recombination hot spots in the non-coding regions.

Reply 1.4: We understand the reviewer's concern, and we did in fact use a rather conservative approach to our scaffolding to avoid generating *in silico* chimeras as much as our data allowed. In this work, two contigs were linked together only when their scaffolding was supported solely by 'proper' read pairs (i.e., mapped paired-end reads with the correct 'inwards' orientation, →←, and they were separated by a reasonable distance expected based on the library preparation or other read pairs), and all supporting read pairs must have also supported only such the scaffolding, and not any other scaffoldings; otherwise, the contigs would have been left unscaffolded.

For example, the contig HS04001_10, which was identified as a non-coding sequence assembled from the dataset HS04001, was not scaffolded to any other contigs as the back read mapping suggested that the contig could be linked to multiple (anellovirus) sequences with proper read pairs, namely HS04001_5, HS04001_8, and HS04001_9.

After scaffolding, we further evaluated the scaffolded sequences by screening for the presence of split reads (i.e., individual reads that could be split into multiple subsegments and mapped to multiple non-contiguous regions). Such reads could be used as an indicator of chimeric sequences and could be relatively easily identified by searching for mapped reads with the flag “2048” using SAMtools view. Our analysis did not identify any such split reads. Furthermore, distances between read pairs supporting our manual scaffoldings were computed, and compared to those of other read pairs in the same datasets. None were found to have unexpected distances, supporting that all read pairs supporting scaffoldings were indeed proper read pairs.

With all of these results, we believe that our scaffolding procedure was justified.

We have now further explained our scaffolding procedure in the Methods section.

Line 143-160: Anellovirus contigs were manually scaffolded based on paired-end read linkages deduced from the read mapping information. In order to avoid generating *in silico* chimeric sequences as much as our data allowed, two contigs were linked together only when their scaffolding was supported solely by ‘proper’ read pairs (i.e., mapped paired-end reads showing the correct ‘inwards’ orientation, $\rightarrow\leftarrow$, and they were separated by a reasonable distance expected based on the library preparation or other read pairs), and all supporting read pairs must have also supported only such the scaffolding, and not any other scaffoldings; otherwise, the contigs would have been left unscaffolded. Potential chimeric

sequences among scaffolds were further checked by searching for 'split' reads in the back-mapping read alignments (i.e., individual reads that could be split into multiple subsegments and mapped to multiple non-contiguous regions) by using SAMtools (25); but no split reads were detected. For scaffolds containing sequence gaps, gap sizes were estimated by aligning the scaffolds to their closest matches in the NCBI nt database. Distances between read pairs supporting our manual scaffoldings were computed, and compared to those of other read pairs in the same datasets. None were found to have unexpected distances, supporting that all read pairs supporting scaffoldings were indeed proper read pairs.~~For scaffolds containing sequence gaps, gap sizes were estimated by aligning the scaffolds to their closest matches in the NCBI nt database.~~

Comment 1.5: 5) All the partial and complete genomes are only available in the supplementary data as a fasta file. Why not deposit these in GenBank using the third part annotations set up. As it stands none of these sequences will be database searchable and with no GenBank accession numbers, they will not get classified.

Reply 1.5: We have now uploaded our sequences to the International Nucleotide Sequence Database Collaboration (INSDC) via the European Nucleotide Archive (ENA), and are now accessible through the Project accession number (BioProject) PRJEB89271. We have now added this information under the section "Data Availability Statement".

Line 589-592: The anellovirus sequences generated in this work are publicly available in the European Nucleotide Archive under the Project accession number PRJNA1054479 (<https://www.ebi.ac.uk/ena/browser/view/PRJEB89271>).~~The anellovirus sequences discovered in this work are available in the~~ **Supplementary Data**

Reviewer #2:

The manuscript investigates the diversity of anelloviruses in Thai individuals by analyzing 1,175 whole genome sequencing (WGS) datasets. The authors identified 511 anellovirus genomes, including 77 complete and 434 partial genomes, spanning seven genera. Notably, four of these genera- Heterotorquevirus, Lamedtorquevirus, Samektorquevirus, and Yodtorquevirus- have not previously been reported in Thailand. Through ORF1-based taxonomic and phylogenetic analyses, the authors propose 33 potentially novel species and discuss inconsistencies in the current ICTV classification system, particularly due to the influence of alignment gaps on pairwise similarity measures.

The manuscript presents high-quality, timely research with substantial potential impact. However, several claims- particularly those related to species classification, evolutionary interpretation, and the degree of novelty- require more cautious framing and further clarification. I recommend major revisions focused on methodological clarification, improved comparative context, and more balanced discussion of results and limitations.

Reply: We thank the reviewer for evaluating our work and constructive comments. Specific points are addressed below.

Comment 2.1: Line 24-26: The claim of 77 complete and 434 partial genomes from 149 samples requires clearer explanation of the criteria used to define "completeness." Was completeness determined solely by CheckV, or were other factors such as circularity or presence of conserved genomic motifs also considered?

Reply 2.1: We appreciate the reviewer's comment, and understand their concern (which was also raised by **Reviewer #1; Comment 1.3**). In this work, we actually only used CheckV as a preliminary tool to assess the quality of our sequence assemblies. To determine the genome completeness, we evaluated the completeness of the *orf1* gene, and looked for the presence of terminal redundancy, and the known conserved untranslated region (UTR) located upstream of the *orf1* gene.

We apologise that this was unclear in our original manuscript, and we have now revised the Abstract, Methods, and Results sections to better describe the criteria used to define genome completeness.

Line 26-30: Our analyses uncovered 77 complete and 434 partial anellovirus genomes from 149 samples (12.68%), detected anellovirus sequences in 149 datasets (12.68%), uncovering 434 partial anellovirus sequences, and 77 complete genome sequences, characterised by the presence of terminal redundancy, complete *orf1*, and the conserved untranslated region upstream of the *orf1* gene.

Line 163-166: Sequence quality was assessed again. Finally, sequence quality was assessed using CheckV version 1.0.1 (20) with the database version 1.5, and, finally, genome sequence completeness was determined based on the presence of terminal redundancy, complete *orf1*, and the conserved untranslated region (UTR) located upstream of the *orf1* gene.

Line 295-298: Characterisation of ~~Among~~ the 511 anellovirus genome assemblies revealed that, 77 of them had the complete *orf1* gene sequence, identical sequences at both contig ends, and the conserved UTR upstream of the *orf1* gene, and, thus, they were deemed to be complete circular genomes of anelloviruses.

Comment 2.2: Lines 85-87: The manuscript does not mention the inclusion of negative controls or contamination assessment. Given that the WGS data were originally generated for human genomic studies, the possibility of detecting false positives or host-derived sequences mimicking viral content must be addressed.

Reply 2.2: In studies designed to test specific hypotheses, for example, potential associations between anelloviruses and particular clinical, biological, or environmental factors, we agree that the inclusion of negative controls and thorough contamination assessment is essential to avoid false conclusions. However, we would like to clarify that our study is descriptive in nature, aiming solely to survey anellovirus sequences in whole genome sequencing data from Thai individuals (contaminants or otherwise).

That said, our analysis did not detect identical anellovirus sequences across multiple datasets, which supports that the sequences we found are unlikely to be the result of systematic laboratory reagent contamination. Furthermore, reciprocal BLASTN searches consistently indicated that the identified sequences showed the highest similarity to known anellovirus sequences, and not to sequences of human (or other cellular organisms) in the NCBI nt database. While we cannot entirely rule out the possibility that (some of) these sequences may in fact be of human origin (and were misidentified as anelloviruses), our results suggest that this is highly unlikely.

Nonetheless, we appreciate the reviewer's concern, and have now addressed and discussed this point in our revised manuscript.

Line 460-478: [O]ur results demonstrate that anellovirus sequences can be unexpectedly found in human sequencing data, even when such if the datasets are not specifically generated for virus discovery/sequence detection specifically. This highlights the potential of human sequencing data as a valuable resource treasure trove for uncovering unseen anellovirus diversity. However, datasets not originally designed for virus discovery, such as ones analysed here, may lack appropriate negative/positive controls to detect viral sequence contaminants or to distinguish genuine viral sequences from non-viral sequences but which exhibit detectable similarity to known viral genomes. It is therefore recommended that studies aiming to test specific hypotheses, such as potential associations between anelloviruses and clinical, biological, or environmental factors, should carefully validate sequence data and rigorously assess and interpret results, particularly when using viral sequences derived from datasets not originally intended for virome studies. That said, our analysis did not detect identical anellovirus sequences across multiple datasets, which suggests that our sequences are unlikely to be the result of systematic laboratory reagent contamination. Furthermore, reciprocal BLASTN searches consistently showed that our sequences exhibited the highest similarity to known anelloviruses rather than to human (or other cellular) sequences, suggesting that they are unlikely to be misidentified host-derived fragments. Taken together, these support the validity of our newly identified anellovirus sequences, and suggest that they may be suitable for use in future studies, at least those focusing on investigation of anellovirus diversity.

Comment 2.3: Lines 87-95 and 212-217: The presence of very long contigs (12-121 kb), far exceeding typical anellovirus genome sizes (~2-4 kb), raises concerns about misassembly or chimeric artifacts. Although the manuscript notes these are likely misassembled, it is unclear how many were ultimately excluded from the analysis and based on what threshold. Please clearly state the filtering strategy for removing implausible contigs.

Reply 2.3: We apologise for not clearly stating how many of the very long contigs were excluded from our analyses, and on what basis. Our reciprocal BLASTN analysis suggested that all of these very long contigs were in fact false positives, and as such all of them were excluded from downstream analysis. We have now clarified this in the revised manuscript.

Line 267-280: ~~[W]e noted that 10 were very long, ranging from between 12.67 to 121.97 kilobases and long-harboured 7-93 ORFs. These lengths are much longer than a typical anellovirus genome (2-4 kilobases), and none of the ORFs were identified as viral protein-encoding genes. These unusually long putative anellovirus sequences also received CheckV “Not-determined” (n=9) and “Low-quality” (n=1) quality score. Altogether, these observations suggest that these~~ These sequences were likely misannotated or misassembled contigs.

Next, reciprocal BLAST analysis was performed. 622 sequences (622/700=88.86%) had anellovirus sequences as their BLAST best-hits being anellovirus sequences, and, thus, were therefore deemed true positives (Fig. 2). The remaining 78 sequences did not return an anellovirus sequence as their best hit (11.14%), including all the unusually long sequences mentioned earlier. These sequences ~~rest~~ were thereby identified as false positives, and were excluded from all subsequent further analyses downstream. The majority (n=64; 82.05%) of the excluded sequences were of the CheckV “Not-determined” quality. As a result, the number of anellovirus sequences-positive samples fell from 178 to 149, 12.68% of the total samples.

Comment 2.4: Lines 128-151 and 258-261: Genus assignment is limited to sequences with ORF1 {greater than or equal to}900 bp, which leaves 299 of the 511 genomes unclassified. The authors should clarify whether any supplementary classification methods were considered for these partial genomes, such as using other ORFs or UTR motifs, or if they were entirely excluded from taxonomic analysis.

Reply 2.4: We appreciate and understand the reviewer’s concern. However, the current classification scheme endorsed by the ICTV *Anelloviridae* Study Group relies exclusively on the analysis of ORF1 protein sequences. In line with this practice, we decided to limit our genus-level classification to only contigs harbouring *orf1* gene sequence of at least 900 bases. As the reviewer correctly noted, this meant that 299 of the 511 sequences (58.51%) were excluded from taxonomic assignment, and we did not use any supplementary classification approaches (e.g., using other ORFs or UTR motifs). This information has now been clarified in the revised manuscript.

Line 193-200: ~~To~~ We assigned the discovered anellovirus sequences into their respective genera based on *orf1* sequence analysis. ORFs with a minimum at least length of 900 bases were extracted from the assembled sequences by using ORFfinder, and they were searched ~~them~~ against the curated reference *orf1* database using ~~with~~ BLASTN using ~~with~~ the following parameters ~~option~~ “-word_size 8 -reward 1 -penalty -1 -gapopen 4 -gapextend 1 -qcov_hsp_perc 50”. A sequence was assigned to the genus of its BLASTN best-hit if they showed 60% identity or greater. Assembled sequences lacking an *orf1* sequence of at least 900 bases were not included in the taxonomic analysis.

Line 303-310: Among the 511 anellovirus sequences discovered, 212 sequences (41.49%) were found to contain substantially long *orf1* sequences (≥900 bases), allowing them to be taxonomically assigned at the genus level using *orf1*. The remaining 299 sequences (58.51%), which either lack *orf1* or had shorter *orf1* sequences, were excluded from taxonomic classification. To this end, the gene sequences were

extracted and searched against ~~a~~the curated taxonomically-annotated reference anellovirus *orf1* database by using BLASTN. Their taxonomic group was identified based on their BLASTN best-hit.

Comment 2.5: Line 217-221: Please specify whether the anomalously long sequences mentioned were completely excluded from downstream analysis. The current language leaves ambiguity regarding whether some might have been retained or mistakenly used.

Reply 2.5: As noted in our response to **Comment 2.3**, all unusually long contigs were excluded from downstream analyses. We apologise again that this was unclear, and have now rewritten this section to improve the clarity.

Line 273-280: Next, reciprocal BLAST analysis was performed. 622 sequences (622/700=88.86%) had anellovirus sequences as their BLAST best-hits~~being anellovirus sequences, and, thus, were therefore~~ deemed true positives (**Fig. 2**). The remaining 78 sequences did not return an anellovirus sequence as their best hit (11.14%), including all the unusually long sequences mentioned earlier. These sequences~~rest~~ were thereby identified as false positives, and were excluded from all subsequent ~~further analyses~~ downstream. ~~The majority (n=64; 82.05%) of the excluded sequences were of the CheckV “Not-determined” quality. As a result, the number of anellovirus sequences-positive samples fell from 178 to 149, 12.68% of the total samples.~~

Comment 2.6: Lines 407-409: While the limitations of short-read sequencing are briefly acknowledged, the discussion would benefit from a more explicit recommendation for the use of long-read sequencing technologies (e.g., Oxford Nanopore, PacBio) in future studies to improve genome reconstruction and ORF validation.

Reply 2.6: We thank the reviewer for this insightful suggestion, and we have now discussed this point in our manuscript.

Line 446-451: This large number of fragmented genome sequences were perhaps expected given that we were limited ~~to~~ only short-read sequence data. To mitigate this potential issue, future metagenomic studies focusing on viral sequence discovery are therefore recommended to incorporate long-read sequencing technologies (e.g., Oxford Nanopore, PacBio) alongside short-read methods to improve genome reconstruction and ORF validation.

Comment 2.7: Line 412-414: The claim that Heterotorquevirus, Lamedtorquevirus, Samektorquevirus, and Yodtorquevirus are newly reported in Thailand is supported. However, it would significantly strengthen the impact of this finding to include a summary comparison table of previously reported genera in Thailand (based on references 6-15) versus those identified in this study.

Reply 2.7: We thank the reviewer for this helpful suggestion. Such a table (**Table 3**) has now been added to our manuscript. Note that we only included studies that identified anelloviruses at the genus level or lower.

Line 479:

Table 3 Anelloviruses reported in humans (*), gibbons (†) and pigs (‡) from Thailand. This table includes only studies that report the virus at the genus level or below.

Genus	Source							
	This study*	(6)*	(7)*	(8)*	(9)*	(14)*	(10)†	(11)‡
Alphatorquevirus	✓	✓	✓	✓	✓	✓	✓	
Betatorquevirus	✓					✓		
Gammatorquevirus	✓					✓		
Hetorquevirus	✓							
Iotatorquevirus								✓
Lamedtorquevirus	✓							
Samektorquevirus	✓							
Yodtorquevirus	✓							

Comment 2.8: Line 416-419: No virome controls, spike-ins, or mock samples are mentioned. A brief discussion on the potential for contamination, especially since the datasets were not intended for virome studies, is necessary to strengthen the reliability of the viral detection claims.

Reply 2.8: We understand the reviewer’s concern, and have now discussed this limitation in our manuscript. (See a related response to **Reply 2.2**).

Line 464-478: However, datasets not originally designed for virus discovery, such as ones analysed here, may lack appropriate negative/positive controls to detect viral sequence contaminants or to distinguish genuine viral sequences from non-viral sequences but which exhibit detectable similarity to known viral genomes. It is therefore recommended that studies aiming to test specific hypotheses, such as potential associations between anelloviruses and clinical, biological, or environmental factors, should carefully validate sequence data and rigorously assess and interpret results, particularly when using viral sequences derived from datasets not originally intended for virome studies. That said, our analysis did not detect identical anellovirus sequences across multiple datasets, which suggests that our sequences are unlikely to be the result of systematic laboratory reagent contamination. Furthermore, reciprocal BLASTN searches consistently showed that our sequences exhibited the highest similarity to known anelloviruses rather than to human (or other cellular) sequences, suggesting that they are unlikely to be misidentified host-derived fragments. Taken together, these support the validity of our newly identified anellovirus sequences, and suggest that they may be suitable for use in future studies, at least those focusing on investigation of anellovirus diversity.

Comment 2.9: Line 419: The use of the phrase "treasure trove" is non-academic and should be replaced with more neutral language such as "valuable dataset" or "rich source."

Reply 2.9: We thank the reviewer for pointing this out. The phrase “treasure trove” have been replaced with “a valuable resource”.

Line 462-464: This highlights the potential of human sequencing data as a valuable resource ~~treasure trove~~ for uncovering unseen anellovirus diversity.

Comment 2.10: Line 423-426: The brief mention of co-infection with multiple anelloviruses should be expanded. The authors are encouraged to include a discussion of the potential immunological or clinical implications of co-infection, referencing recent literature on host immune modulation.

Reply 2.10: We thank the reviewer for this insightful suggestion. While several studies have now reported associations between anellovirus abundance and host immune status or disease severity in humans, the biological implications of co-infection with multiple anellovirus lineages still remain largely understudied indeed. We have now expanded this point in the revised manuscript.

Line 482-496: Recent studies analysing HTS NGS-data have reported multiple distinct lineages of anelloviruses co-existing within single individuals (36–38, 41). In line with these findings, we found identified 24 out of 61 samples in our dataset from which complete *orf1* sequences could be recovered to contain contained multiple distinct anellovirus sequences (i.e., belonging to different strict clusters). These could be a result of sequential infections, or a single “bulk transmission” into the persons, or a combination of both. Indeed, a recent study showed that blood transfusion can introduce multiple lineages of anelloviruses to recipients (36). Given that a notable proportion of our data (24/61 = 39.34%) appeared to contain multiple lineages of anelloviruses, it may be of interest to explore the potential immunological or clinical implications of such co-infections. Although several previous studies have already linked overall anellovirus load with host immune status, cancer, and the severity of several diseases in humans (3, 40, 42–44), anelloviruses are often treated as a single viral population, and the effects of co-infecting lineages have yet remained understudied. Further functional and systemic investigations are needed to shed more light on the roles of anellovirus diversity and co-infection in human health and disease.

Comment 2.11: Line 440-441: The statement that these findings “warrant further systematic screening for novel anellovirus discovery” is too strong in the absence of any functional validation. Please consider rephrasing this to suggest that the results “support the potential utility” of WGS data for future virome research.

Reply 2.11: We understand the reviewer’s concern and agree with them. We have now removed the original sentence from our manuscript, and revised the text accordingly.

Line 515-519: These findings illustrate a considerable and largely considerably large and unexplored anellovirus diversity in Thailand, and at the same time highlight the potential of human sequencing data as a useful resource for future virome research and virus discoverywarranting further systematic screening for novel anellovirus discovery.

Comment 2.12: Lines 449-454: The claim that the classification scheme is unstable due to reference dataset dependency is overstated. While the study shows variability around the 69% similarity threshold, this does not in itself prove instability across the classification system. A more measured phrasing is recommended.

Reply 2.12: We thank the reviewer for pointing this out, and agree with them. We have now revised the sentence to be more specific about sequences showing pairwise *orf1* sequence similarity values near the 69% threshold.

Line 528-533: However, as previously shown (4), while there indeed exists a trough around at ~69% pairwise *orf1* sequence identity, it the trough is not completely empty (i.e., there exist are pairs of viruses that show *orf1* sequence similarity around 68-70%). This, in turn, can cause suggests that the current classification scheme can to be somewhat dependent sensitive to on the reference datasets

used, and to some extent unstable particularly when involves anelloviruses showing pairwise *orf1* sequence similarity values near the 69% threshold.

Comment 2.13: Line 454-460: The idea that increased sampling could collapse distinct species into a single mega-cluster under the current classification criteria is an interesting hypothesis but remains speculative. Please qualify this as a theoretical concern rather than a demonstrated outcome.

Reply 2.13: We thank the reviewer for pointing this out. The sentence has now been revised to explicitly state that this is only a theoretical concern.

Line 540-549: ~~As we continue to investigate anellovirus diversity from more and more underexplored unexplored geographical regions and human populations, the observed gap in the virus genetic diversity will likely diminish and the trough of the current species demarcation might become less and less pronounced. In theory, it is therefore In an extreme scenario, it might even be possible that, with continual sequencing efforts and novel anellovirus discoveries, a large and dense network of anellovirus sequences might emerge, unifying many species of anelloviruses together into one under the current pairwise similarity-based classification scheme. While this is entirely hypothetical and speculative, it highlights a possible theoretical limitation of the current classification scheme in the face of increasing sampling.~~

Comment 2.14: Line 483-485: The suggestion that all classification results should be cross-validated with phylogenetic analysis is valid. However, the phrasing implies a frequent conflict between methods. Please soften this claim unless the manuscript quantifies the frequency and impact of such discrepancies.

Reply 2.14: We thank the reviewer for pointing this out, and the text has been revised accordingly. The discussion has now been revised to focus specifically on classification results derived from alignments with high gap proportions.

Line 565-570: ~~Thus, relationships derived based on overall similarity calculated from pairwise alignments with high gap proportions must should be carefully examined and cross-checked. With the current classification scheme, it is important to cross-check the results with phylogenetic analysis to ensure that the resultant classification is consistent with the evolutionary history of the viruses, especially when the similarity values are calculated from pairwise alignments with high gap proportions.~~

Comment 2.15: Figures 5 & 6: These figures are important for interpreting species boundaries and novel clusters. However, the legends should be expanded to define what "strict" and "relaxed" clusters represent and how these classifications were applied in the study. For readers unfamiliar with similarity network analysis, this added context would improve accessibility.

Reply 2.15: We appreciate the reviewer's comment. In addition to **Figures 5** and **6**, we would like to note that the legend of **Figure 1** describes the species assignment procedure based on sequence similarity network analysis.

To improve accessibility for readers who may not be familiar with this method, we have now revised the legends of **Figures 1, 5, and 6** to better define "strict" and "relaxed" linkages, and explain how these

linkages were applied in the analysis and species assignment. In **Figures 5** and **6**, readers are referred to **Figure 1**'s legend for further details, in order to minimise redundancy in the manuscript.

Line 815-853: Figure 1 Anellovirus sequence-species assignment based on pairwise sequence similarity network analysis lower- and upper-bound sequence similarities. Sequences are represented by colour-coded vertices. Grey edges denote '*strict linkages*', indicating that linked sequences show both lower- and upper-bound similarity >69%. Green edges denote '*relaxed linkages*', indicating that linked sequences show upper-bound similarity >69% but lower-bound similarity ≤69%. Species assignment is based on sequence clustering patterns formed using these two linkage types. In this hypothetical network, sequence Sequence-A is classified as a member of a potentially novel anellovirus species because it belongs to is a member of a '*relaxed cluster*' lacking any ICTV exemplars (i.e., sequence A still does not show similarity above the threshold to any ICTV exemplars even when intermediate sequences are considered via relaxed linkages). Sequence B is classified to species *M* because it belongs to a '*strict cluster*' exclusively containing with the only one ICTV exemplar of species *M* sequence, and the its relaxed cluster containing sequence B does not contain any additional other ICTV exemplars (i.e., even with only strict linkages, sequence B still shows similarity above the threshold exclusively to the species *M* exemplar even when intermediate sequences are considered, and including relaxed linkages does not change that). For sequence Sequence-C-, while it is linked to one, and only one, has an undetermined status because there is an ICTV exemplar, establishing this linkage requires at least one relaxed linkage in the same relaxed cluster, and they do not belong to but they are not in the same strict cluster, so it is assigned an undetermined status (i.e., the assignment depends on linkage type used, and thus remains unresolved). Sequence D is also assigned also has an undetermined status because it is part a member of a relaxed cluster with multiple ICTV exemplars (i.e., due to the possibility of belonging to a multi-species cluster, its assignment hence remains unresolved here). Sequences are represented by colour-coded vertices. A grey edge represents a strict linkage (>69% lower- and therefore upper-bound sequence similarity), while a green edge represents a relaxed linkage (>69% upper-bound but ≤69% lower-bound sequence similarity).

Line 854-876: Figure 5 *orf1* sequence similarity networks and phylogenies of (A) alphatorqueviruses, (B) betatorqueviruses, (C) gammatorqueviruses, and (D) heteorqueviruses. Panels (A) to (D) show *orf1* sequence similarity networks for alphatorqueviruses, betatorqueviruses, gammatorqueviruses, and heteorqueviruses, respectively. Orange, blue, and grey vertices represent complete *orf1* sequences from the present study, complete *orf1* sequences of from the ICTV anellovirus exemplars, and of from non-exemplar contextualising reference sequences, respectively. The vertices representing of sequences from this study and from ICTV anellovirus exemplars are enlarged for visual emphasis. Grey edges denote '*strict linkages*', indicating that linked sequences show both lower- and upper-bound similarity >69%. Green edges denote '*relaxed linkages*', indicating that linked sequences show upper-bound similarity >69% but lower-bound similarity ≤69%. Sequences are linked by grey edges if they have lower-bound sequence similarities > 69%, and by green edges if they have upper bound sequence similarity > 69% (but not lower bound sequence similarity). Relaxed clusters containing Thai anellovirus sequences without lacking an ICTV exemplars (i.e., potentially novel species) are outlined with highlighted with a purple lines border, while clusters exclusively comprising containing Thai anellovirus sequences are highlighted with a outlined with red lines border. Panels (E) to (H) show phylogenetic trees of ORF1 protein sequences for Phylogenetic analysis of ORF1 protein sequences of (E) alphatorqueviruses (reconstructed were conducted using the LG+F+R8 model), (F) betatorqueviruses using the (LG+F+R10) model, (G) gammatorqueviruses using the (LG+F+R10) model, and (H) heteorqueviruses using the (LG+F+[G4] model, respectively. Green and grey curved lines denote relaxed and strict linkages, respectively. Relaxed or strict similarity links are indicated on the trees by green and grey curve lines,

respectively. The ICTV mutorquevirus exemplars were used to root the trees and are not shown. Trees are rooted using ICTV mutorquevirus exemplar sequences (not shown). Orange and blue solid circles indicate sequences from this study and ICTV exemplars, respectively; undecorated tips represent non-exemplar contextualising sequences. Tips marked with orange and blue circles represent sequences found in the present study and those of the ICTV exemplars, respectively. Undecorated tips represent non-exemplar reference sequences.

Line 878-885: Figure 6 Species assignments of Thai anellovirus sequences discovered in this study Numbers of Thai anellovirus sequences discovered in this study identified as belonging to known species (A) and potentially novel species (B). Classification of Thai anellovirus sequences generated in this study was conducted based on complete *orf1* sequence similarity network analyses (see Figure 5), using the classification scheme described in Figure 1. Panels show the number of sequences of (A) known species, (B) potentially novel species, and (C) having undetermined species status. Each stacked bar in panels (A) and (B) represents an individual virus cluster/species. Each stacked bar represents an individual virus species as determined by sequence similarity network analyses (Fig. 5).

Comment 2.16: Table S2: While valuable, this supplementary table would be more informative if key data-such as the number of sequences assigned to known species, novel species, or with undetermined status-were summarized within the main text.

Reply 2.16: We thank the reviewer for the suggestion. To improve the readability, we have now revised Figure 6 to show the number of sequences assigned to known species, potentially novel species, and those with undetermined species assignment status.

Line 878-885: Figure 6 Species assignments of Thai anellovirus sequences discovered in this study~~Numbers of Thai anellovirus sequences discovered in this study identified as belonging to known species (A) and potentially novel species (B).~~ Classification of Thai anellovirus sequences generated in this study was conducted based on complete *orf1* sequence similarity network analyses (see **Figure 5**), using the classification scheme described in **Figure 1**. Panels show the number of sequences of **(A)** known species, **(B)** potentially novel species, and **(C)** having undetermined species status. Each stacked bar in panels **(A)** and **(B)** represents an individual virus cluster/species. Each stacked bar represents an individual virus species as determined by sequence similarity network analyses (**Fig. 5**).

The summary of this information has also been now mentioned in the text.

Line 507-509: With sequence similarity network analysis, we identified 55 of our anellovirus sequences as ~~potentially~~ belonging to 33 potentially new species, 32 sequences belonging to 13 known species, and 47 sequences with undetermined status.

Comment 2.17: Additional Suggestion: The manuscript would benefit from a visual or tabular summary of newly identified species by genus and geographic origin (if such metadata exist). This would help highlight the uniqueness of the Thai anellovirus dataset within the global context

Reply 2.17: We thank the reviewer for this constructive suggestion. In response, we further explored the uniqueness of Thai anelloviruses by comparing the relaxed clustering profiles of anelloviruses from Thailand to those from other countries, using the probability Jaccard Index (J_p). The results are reported in the revised manuscript, along with a new figure summarising the findings.

Line 227-238: Similarity between anellovirus profiles of Thailand and other countries

To quantify the similarities between the anellovirus profile of Thailand and those of other countries, we employed pairwise probability Jaccard scores (J_p) (31). For two probability distributions $X = (x_1, x_2, \dots, x_n)$ and $Y = (y_1, y_2, \dots, y_n)$, where $x_i, y_i \in [0,1]$ for all x_i, y_i and $\sum_i x_i = 1$, and $\sum_i y_i = 1$, their J_p score can be computed as follows:

$$J_p(X, Y) = \sum_{x_i \neq 0, y_i \neq 0} \frac{1}{\sum_j \max\left(\frac{x_j}{x_i}, \frac{y_j}{y_i}\right)} \quad (3)$$

The country of origin for each sequence was extracted from the “geo_loc_name” field in the NCBI nt database, and sequences lacking this information (524/21,295=2.46%) were excluded from the analysis. For anellovirus profile construction, relaxed clustering results were used, and non cluster-representative sequences from the NCBI nt database (n=17,451) were re-assigned back to the relaxed clusters (n=368) of their representative sequences for the probability (i.e., frequency) calculation. J_p values were then computed between Thailand’s profile and those of all other countries.

Line 376-388: Thai anellovirus diversity in the global context

Based on the relaxed clustering networks (supplemented with 17,451 non cluster-representative sequences from the NCBI nt database), we compared the anellovirus profiles of Thailand against those of 31 other countries (n clusters=368) by using the probability Jaccard Index J_p (31). We found that 22 countries across Africa (3 countries), Americas (3 countries), Asia (7 countries), and Europe (9 countries) shared at least one relaxed cluster with Thailand, while the remaining nine countries did not (**Fig. 7** and **Table S3**). Switzerland showed the greatest similarity to Thailand ($J_p = 0.4115$), followed by China ($J_p = 0.3641$), and Spain ($J_p = 0.3638$). Interestingly, despite their geographic proximity, Malaysia ($J_p = 0.0411$) and Vietnam ($J_p = 0.0217$) showed low similarity to Thailand (**Fig. 7**). These results support that Thailand

does share anelloviruses with various countries; however, within this dataset, the overall composition of the viruses appears distinct, and does not necessarily correlate with geographic proximity.

Line 887-891: Figure 7 Pairwise probability Jaccard score (J_p) between anellovirus profiles of Thailand and each of other countries. The map shows J_p values, comparing the anellovirus profile of Thailand with those of other countries derived based on relaxed sequence similarity clusters supplemented with 17,451 non cluster-representative sequences from the NCBI nt database. Colour scale indicates J_p values, ranging from 1 (high similarity; blue) to 0 (no shared clusters; red). Countries shown in white are those without anellovirus sequence data in our dataset.

Comment 2.18: Furthermore, the importance section should be revised as it is not the summary of your findings or repeating the information within the abstract.

Reply 2.17: The Importance section has now been revised to avoid repeating information already presented in the abstract.

Line 47-68: Anelloviruses are widespread in humans, yet their diversity remains poorly characterised in many regions, including Thailand. Here, we demonstrate that human sequencing datasets, originally generated without the intention for virome research, can be effectively mined for anellovirus sequences, including complete genomes. Our findings reveal a substantial number of previously unreported anelloviruses in Thailand, significantly expanding the known diversity of the virus. We also highlight potential limitations of the current anellovirus species classification scheme, which is based on pairwise orf1 sequence similarity analysis with a hard threshold cut-off at 69%. Our results reveal that the current scheme can sometimes yield taxonomic groupings that are inconsistent with phylogenetic relationships, particularly when significant alignment gaps are present. Overall, our results show that existing human sequencing data can be effectively repurposed for virus discovery research, and suggest the need for more robust and phylogenetically informed classification frameworks as viral sequence databases

continue to expand. The current understanding of anelloviruses in Thailand is limited. This study reports the largest collection of anellovirus sequences from Thailand, comprising 434 partial genomes and the first 77 complete genomes of anelloviruses mined from 1,175 whole genome sequencing datasets of Thai human individuals. Consistent with previous studies, our analyses revealed the presence of alpha-, beta-, and gammatorqueviruses in the Thai population, but have now also expanded the documented viral diversity to include he-, lamed-, samek-, and yodtorqueviruses, with a total of 33 potentially novel species identified. These findings illustrate the potential use of human sequencing data as a valuable resource for virus discovery and highlight the vast and largely unexplored anellovirus diversity in Thailand. The current classification scheme of anellovirus species is also discussed under the light of the expanding viral diversity.

Re: Spectrum00866-25R1 (Discovery of diverse anellovirus sequences in Thai human sequencing data)

Dear Dr. Pakorn Aiewsakun:

Thank you for the privilege of reviewing your work. Below you will find my comments, instructions from the Spectrum editorial office, and the reviewer comments.

As you will see, your manuscript has progressed to the final stage of consideration for publication in the journal. Prior to formal acceptance, we kindly request that you provide valid accession numbers (not project identifiers) for all anelloviral sequences generated in this study, which must be deposited in an internationally recognized public repository. This information is essential for ensuring scientific reproducibility and enabling readers to access the sequences supporting your findings. Please confirm the availability of these accession numbers in your revised submission.

Revision Guidelines

Sincerely,
Biao He
Editor
Microbiology Spectrum

Reviewer #2 (Public repository details (Required)):

The data use in the study should be published in public database.

Reviewer #2 (Comments for the Author):

Authors have addressed my particular concerns.

Point-by-point response

Editor:

Comment 1.1: As you will see, your manuscript has progressed to the final stage of consideration for publication in the journal. Prior to formal acceptance, we kindly request that you provide valid accession numbers (not project identifiers) for all anelloviral sequences generated in this study, which must be deposited in an internationally recognized public repository. This information is essential for ensuring scientific reproducibility and enabling readers to access the sequences supporting your findings. Please confirm the availability of these accession numbers in your revised submission.

Reply 1.1: We have now provided valid accession numbers for all anelloviral sequences generated in this study, and all are publicly available in the European Nucleotide Archive. The Data Availability Statement has been revised accordingly to present the information:

Line 556-559: Data Availability Statement

The anellovirus sequences generated in this work are publicly available in the European Nucleotide Archive under with the Project accession numbers OZ258605 to OZ258816 and OZ285112 to OZ285410 PRJNA1054479 (<https://www.ebi.ac.uk/ena/browser/view/PRJEB89271>).

Supplementary Table S2 has also been revised to include accession numbers for all individual anelloviral sequences generated in this study.

Reviewer #2:

Comment 2.1: The data use in the study should be published in public database.

Reply 2.1: We have now provided valid accession numbers for all anelloviral sequences generated in this study, and all are publicly available in the European Nucleotide Archive. The Data Availability Statement has been revised accordingly to present the information:

Line 556-559: Data Availability Statement

The anellovirus sequences generated in this work are publicly available in the European Nucleotide Archive under with the Project accession numbers OZ258605 to OZ258816 and OZ285112 to OZ285410 PRJNA1054479 (<https://www.ebi.ac.uk/ena/browser/view/PRJEB89271>).

Supplementary Table S2 has also been revised to include accession numbers for all individual anelloviral sequences generated in this study.

Comment 2.2: Authors have addressed my particular concerns.

Reply 2.2: Thank you for the feedback. We appreciate all previous constructive comments.

Re: Spectrum00866-25R2 (Discovery of diverse anellovirus sequences in Thai human sequencing data)

Dear Dr. Pakorn Aiewsakun:

Your manuscript has been accepted, and I am forwarding it to the ASM production staff for publication. Your paper will first be checked to make sure all elements meet the technical requirements. ASM staff will contact you if anything needs to be revised before copyediting and production can begin. Otherwise, you will be notified when your proofs are ready to be viewed.

Sincerely,
Biao He
Editor
Microbiology Spectrum